# A cognitive neurogenetic approach to uncovering the structure of executive functions

Junjiao Feng[1,2], Liang Zhang [1], Chunhui Chen [1], Jintao Sheng[1], Zhifang Ye [1], Kanyin Feng[1], Jing Liu[1], Ying Cai[3], Bi Zhu[1], Zhaoxia Yu [4], Chuansheng Chen[5], Qi Dong[1] & Gui Xue [1]✉

One central mission of cognitive neuroscience is to understand the ontology of complex cognitive functions. We addressed this question with a cognitive neurogenetic approach using a large-scale dataset of executive functions (EFs), whole-brain resting-state functional connectivity, and genetic polymorphisms. We found that the bifactor model with common and shifting-specific components not only was parsimonious but also showed maximal dissociations among the EF components at behavioral, neural, and genetic levels. In particular, the genes with enhanced expression in the middle frontal gyrus (MFG) and the subcallosal cingulate gyrus (SCG) showed enrichment for the common and shifting-specific component, respectively. Finally, High-dimensional mediation models further revealed that the functional connectivity patterns significantly mediated the genetic effect on the common EF component. Our study not only reveals insights into the ontology of EFs and their neurogenetic basis, but also provides useful tools to uncover the structure of complex constructs of human cognition.

The ontology of mental constructs serves as the building blocks for our understanding of the human brain and cognition[1]. Yet, as noted by William Uttal[2] "hypothetical psychological constructs are invented ad lib and ad hoc without adequate consideration of the fundamental issue of the very plausibility of the precise definition". One major challenge in uncovering the ontology of complex human cognition is to extract the latent, hypothetical mental constructs from the cognitive tasks designed supposedly to tap them. Although researchers have cautioned about the danger of conflating latent constructs with operational measures[3], it remains a common practice in cognitive neuroscience to equate task with construct[1], resulting in construct impurity[4]. Here, we propose an integrative, data-driven, gene-brain-behavior approach as a framework to discover the ontology of mental constructs.

We focused on the structure of executive functions (EFs), which is particularly interesting and important not only because EFs play a key role in achieving goal-directed behavior[4,5] but also due to their complex structure and the methodological challenges involved. Previous studies have demonstrated that EFs are closely related to many cognitive functions, such as creativity[6,7], intelligence[6,8,9], attention[10], reasoning[11], reading[11], and arithmetic[11]. Furthermore, deficits in EFs have been linked to many mental disorders, such as attention-deficit hyperactivity disorders (ADHD)[12], schizophrenia (SCZ)[13,14], Alzheimer's disease[15], and autism[16]. Therefore, revealing the structure of EFs can contribute to a better understanding of the nature of EFs and their relationship with other cognitive functions, as well as the nature of brain dysfunctions in neurological and psychiatric disorders.

Previous studies have attempted to reveal the structure of mental processes by testing a large sample of subjects with several different tasks[17,18], but the results are mixed. Early studies used confirmatory factor analysis (CFA) to examine the structure of EFs in terms of three core EF components, including inhibiting, updating, and shifting[18,19].

[1]State Key Laboratory of Cognitive Neuroscience and Learning & IDG/McGovern Institute for Brain Research, Beijing Normal University, 100875 Beijing, China. [2]Faculty of Psychology, Tianjin Normal University, 300387 Tianjin, China. [3]Department of Psychology and Behavioral sciences, Zhejiang University, 310017 Hangzhou, China. [4]Department of Statistics, University of California, Irvine, CA 92697, USA. [5]Department of Psychological Science, University of California, Irvine, CA 92697, USA. ✉e-mail: gxue@bnu.edu.cn

However, because the three components are also significantly correlated with one another, later studies concluded that the "Common + Updating-specific + Shifting-specific" bifactor model could better characterize the nature of EFs[20]. These results demonstrate the "unity and diversity" characteristics of EFs[4,18,21].

Moving beyond the pure behavioral model of EFs, researchers have recently argued that we should use biological discoveries to inform the continual development of psychological theories[1]. The underlying logic is that a more biologically plausible model of EFs should have its components rooted in clearly dissociated neural substrates (see Friedman et al.[4] for a review). Thus far, researchers have used three strategies to integrate biological (mainly neural) data into EF models. One strategy is to record participants' brain activity when performing multiple EF tasks and use conjunction and interaction analyses to identify the common and domain-specific neural substrates, respectively[22]. Due to the high cost of scanning many subjects with multiple tasks, existing studies have included few subjects and/or few tasks[22,23], which inevitably led to unreliable results or impurity of EF components[4]. Moreover, the task-evoked approaches also face several other challenges, such as isolating various task components involved in a given cognitive task and identifying the specific brain-behavior associations[1].

The second approach is to use large-scale Meta-Analytic Structure-to-Function Mapping[24], which has been applied to decode a large number of cognitive tasks[25], manually annotated mental processes[26], and various task features[27]. This approach provides a scalable and economical tool for ontology discovery of complex traits and hypothesis generation. Nevertheless, the meta-analytic approach is still under development, and the identification of mental constructs and brain-behavioral mapping still needs further improvement.

The third strategy is to take an individual difference approach by developing a large-scale brain-behavior database and associating individuals' latent component scores of EFs with brain measures. This strategy typically relies on structural MRI[28,29], or resting-state fMRI[30,31], and sometimes task fMRI with a few tasks[32,33], because they are less time consuming and more cost-efficient than scanning all tasks and thus allow for a larger sample size and leave time for more out-of-scanner behavioral tasks. Nevertheless, most studies usually used a small number of behavioral tasks and/or a small sample. More critically, they used neural data only to verify their pre-defined model of mental structure rather than to assess several candidate models.

To overcome the above limitations, we proposed a cognitive neurogenetic approach that integrates genetic, neural, and behavioral data to examine the structure of EFs (Fig. 1). Specifically, we used a large sample of Han Chinese adults ($n = 2110$), a comprehensive battery of cognitive tests (nine different EF tasks), whole-genome scans (a subsample, $n = 1454$), and resting-state neuroimaging data (a subsample, $n = 870$). First, we used confirmatory factor analysis (CFA) to evaluate 12 candidate latent variable models of EFs, which yielded five models with good fitting parameters for subsequent analyses (Fig. 1a). Second, we further confined the five models using resting-state functional connectivity data and connectivity-based predictive model. Specifically, we selected one model whose components could be significantly predicted by nonoverlapping brain connectivity patterns (Fig. 1b). Third, we used meta-analytic data from Neurosynth[24] (https://www.neurosynth.org) to verify the neural results and to identify the neuronal regions associated with each component in the selected model (Fig. 1c). Fourth, we adopted an integrative gene-brain-behavior approach[34–36] to examine the genetic dissociation and enrichment pattern of different EF components (Fig. 1d). Finally, we applied a high-dimensional mediation model[37] to examine the genes-brain-EFs pathway (Fig. 1e). Our study not only identifies a model of EFs that fits the behavioral results well and is supported by neural and genetic evidence, but also provides a cognitive neurogenetic approach that can be applied to examine the structure of other complex traits.

## Results

### Descriptive statistics and bivariate correlations between tasks

To estimate the latent variable models of EFs, our study used nine tasks, including three inhibiting tasks (anti-saccade, stop-signal, and Stroop), three updating tasks (keep track, letter 3-back, and spatial 2-back), and three shifting tasks (number–letter, color–shape, and category switch). Detailed descriptive statistics for the nine EF tasks used in this study are presented in Supplementary Table 1, and their correlations are presented in Supplementary Table 2. Overall, all dependent measures of the nine tasks showed normal distributions, and internal consistency was high for most of the tasks (0.70–0.90), except the Stroop task (0.33), the category switch task (0.51), and the stop-signal task (0.55).

Note that only 1454 out of 2110 participants had data for all the nine tasks, therefore, we used the "pairwise-complete" method when computing the correlation between each pair of variables (see "Methods"). Because the missingness is independent of the unobserved value, the estimates are expected to be unbiased[20,38]. Consistent with existing studies, the correlations among tasks were generally low ($r_{mean} = 0.12$, range: 0.01–0.32), but tasks measuring the same EF component tended to show higher correlations with one another: inhibiting ($r_{mean} = 0.16$, range: 0.10–0.26), updating ($r_{mean} = 0.22$, range: 0.14–0.31), and shifting ($r_{mean} = 0.29$, range: 0.24–0.32). Interestingly, two inhibiting-related tasks (i.e., anti-saccade and stop-signal) were also significantly correlated with the three updating-related tasks (i.e., keep track, letter 3-back, and spatial 2-back) ($r_{mean} = 0.18$, range: 0.16–0.22).

### Testing the candidate models with CFA on the behavioral data

**Model estimation.** To examine the structure of EFs comprehensively, we compared 12 candidate latent variable models based on the various combinations of the three most commonly discussed EF components (i.e., inhibiting [I], updating [U], and shifting [S]). Among them, five are correlated-factors models, and the remaining seven are bifactor models. The five correlated-factors models include the full three-factor model ("I + U + S" model)[18], which assumes no constraints on the three EF components; the one-general-factor model ("G" model) that assumes no separability of the three EF components; and the three two-factor models, which assume that two of the three components are the same and can be combined (i.e., the "shifting = inhibiting" ["S/I + U"] model, the "updating = inhibiting" ["U/I + S"] model, and the "shifting = updating" ["S/U + I"] model).

In addition to the five correlated-factors models, there are seven bifactor models. In the bifactor models used in Friedman and Miyake[4,20], a common EF component is first estimated using correlations among all tasks to capture the unity of EFs. After that, three orthogonal components (i.e., inhibiting-specific, updating-specific, and shifting-specific) are estimated using the remaining correlations among the inhibiting, updating, and shifting tasks, respectively. One advantage of bifactor models is that they allow the estimation of the relationship between other individual differences and the unity and diversity EF components in a more direct way[4]. We evaluated seven bifactor models, including a bifactor three-factor model, which consists of a common EF component and three specific components, i.e., the "common + inhibiting-specific + updating-specific + shifting-specific" model ("C + I + U + S" model); three models with the common EF component and two specific components, i.e., "C + I + S", "C + U + S", and "C + I + U"; and three models with the common EF component and one specific component, i.e., "C + I", "C + S", and "C + U".

The model fit results are provided in Table 1. Five models, including the "I + U + S", "U/I + S", "C + I + S", "C + U + S", and "C + S" models, met the criteria for good fit (CFI > 0.95, SRMR < 0.05, RMSEA < 0.05) (Fig. 2 and see Supplementary Fig. 2 for the remaining seven models that did not fit the data very well). It is worth noting that although the fit indices of the "C + I + U + S" model were also good

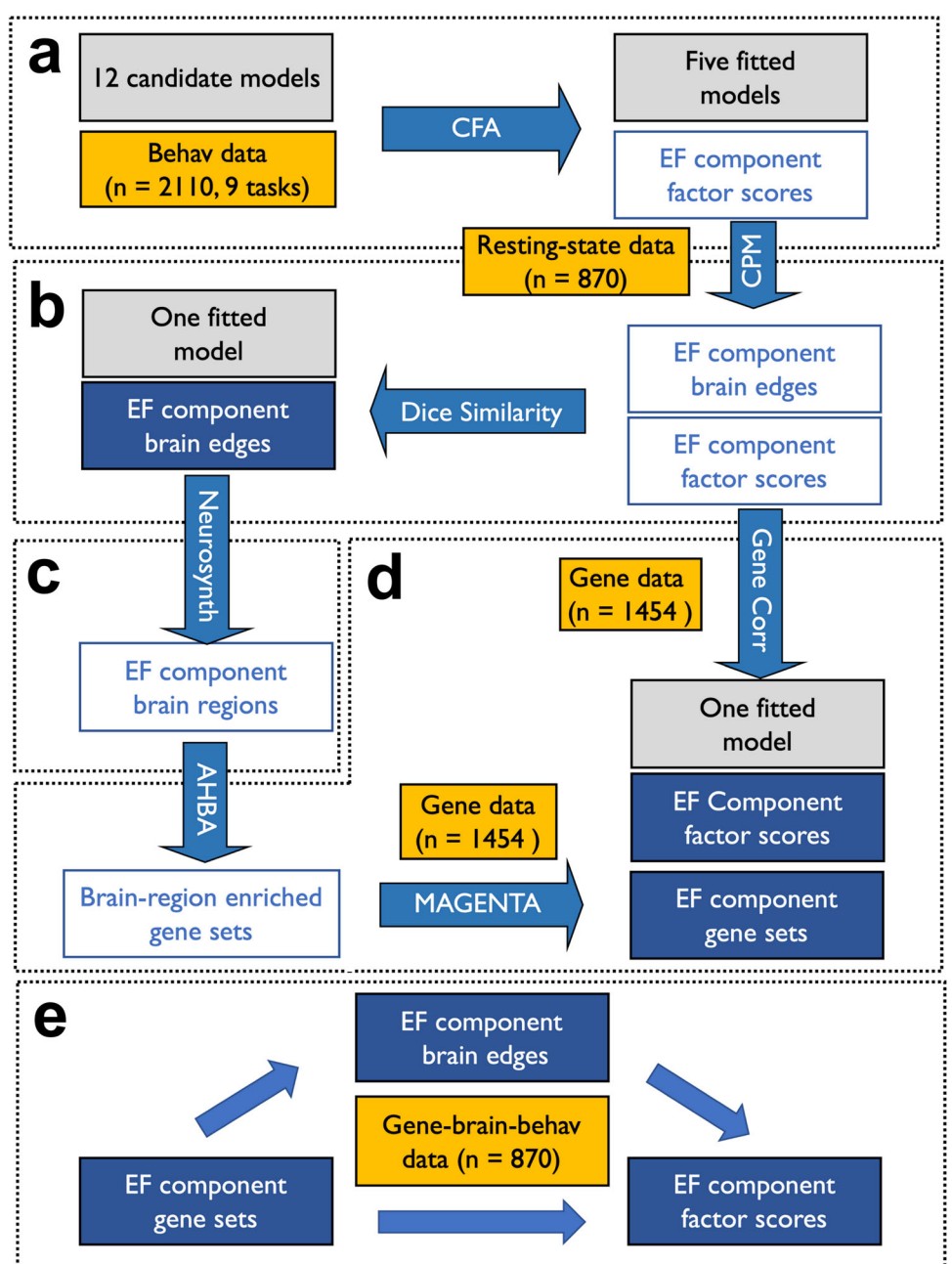

**Fig. 1 | Flowchart of this study. a** Testing the candidate models with CFA on the behavioral data. **b** Using the neural data to constrain the models based on the behavioral data. **c** Combining CPM and Neurosynth to identify brain regions for each component in the selected model. **d** Using the genetic data to constrain the models selected based on behavioral data and to characterize the genetic architecture of EF components based on Allen Human Brain Atlas (AHBA). **e** The gene-brain-behavior pathway for EF components. The yellow boxes show the sample size and data, the gray boxes show the behavioral models, the white and blue boxes show the intermediate products of the gene, brain, and behavioral data in the processing pipeline, and the dark blue boxes show the final products of the gene, brain, and behavioral data used for the pathway analysis. CPM a connectome-based predictive model, MAGENTA Meta-Analysis Gene-set Enrichment of variaNT Associations, AHBA the Allen Human Brain Atlas.

(CFI = 0.99, RMSEA = 0.02, SRMR = 0.02), the tasks' loadings on the inhibiting-specific component (anti-saccade: $P = 0.29$, stop-signal: $P = 0.25$, Stroop: $P = 0.17$) and the updating-specific component (keep track: $P = 0.51$, spatial 2-back: $P = 0.49$, letter 3-back: $P = 0.43$) were not significant. We also re-estimated the EF latent variable models using list-wise deletion ($n = 1454$), and found the same five good-fit models (Supplementary Table 3).

It is worth noting that the "G" model had the poorest fit (CFI = 0.65, RMSEA = 0.08, SRMR = 0.06), indicating the "diversity" of EFs, which is consistent with the previous studies[18]. Meanwhile,

correlations among the three components in the "I + U + S" model were significantly larger than zero, i.e., inhibiting & updating ($r = 0.71$, $P < 1 \times 10^{-3}$, uncorrected), inhibiting & shifting ($r = 0.27$, $P < 1 \times 10^{-3}$, uncorrected), updating & shifting ($r = 0.23$, $P < 1 \times 10^{-3}$, uncorrected). These results support the "unity" pattern of EFs. It should also be emphasized that the correlation between the updating and inhibiting components was much higher than their correlations with the shifting component. In addition, the "U/I + S" model and "C + S" model fit our behavioral data well, which supports the notion that some components of inhibiting and updating abilities are inseparable[39] and that

**Table 1 | Model fit statistics of the 12 EF latent variable models**

| Model | $\chi^2$ | df | CFI | RMSEA | SRMR | AIC | BIC |
|---|---|---|---|---|---|---|---|
| **Correlated-factors models** | | | | | | | |
| **1. I+U+S** | **41.17** | **24** | **0.99** | **0.02** | **0.02** | **48,991** | **49,161** |
| 2. S/I+U | 339.83 | 26 | 0.73 | 0.08 | 0.06 | 49,286 | 49,444 |
| **3. U/I+S** | **74.41** | **26** | **0.96** | **0.03** | **0.02** | **49,021** | **49,179** |
| 4. S/U+I | 402.08 | 26 | 0.68 | 0.08 | 0.06 | 49,349 | 49,507 |
| 5. G | 431.91 | 27 | 0.65 | 0.08 | 0.06 | 49,376 | 49,529 |
| **Bifactor models** | | | | | | | |
| 6. C+I+U+S | 28.45 | 18 | 0.99 | 0.02 | 0.02 | 48,991 | 49,195 |
| **7. C+I+S** | **43.83** | **21** | **0.98** | **0.02** | **0.02** | **49,000** | **49,187** |
| 8. C+I+U | 219.96 | 21 | 0.83 | 0.07 | 0.06 | 49,176 | 49,363 |
| **9. C+U+S** | **30.01** | **21** | **0.99** | **0.01** | **0.02** | **48,987** | **49,173** |
| 10. C+I | 401.81 | 24 | 0.67 | 0.09 | 0.06 | 49,352 | 49,522 |
| **11. C+S** | **73.29** | **24** | **0.96** | **0.03** | **0.02** | **49,024** | **49,193** |
| 12. C+U | 331.92 | 24 | 0.73 | 0.08 | 0.06 | 49,282 | 49,452 |

$\chi^2$ chi-squared statistics, *df* degrees of freedom, *CFI* comparative fit index, *RMSEA* the root-mean-square error of approximation, *SRMR* standardized root-mean square residual, *AIC* Akaike information criterion, *BIC* Bayesian information criterion.
CFI > 0.95 is commonly used as an indication of the adequate fit. Lower values of SRMR and RMSEA indicate better fit, with < 0.05 indicating a good fit. Lower values of AIC and BIC indicate better fit. The good-fit models are indicated in bold, of which the C+U+S model showed the best overall fit.

their shared mechanisms can be explained by the common EF component.

**Model comparisons.** Following previous studies[18,20], to determine which of these five good-fit models had the best fit to our behavioral data, we further used the chi-square ($\chi^2$) difference test to compare nested models[40]. Models are nested when the parameters of one model are a subset of the parameters of another model. In our study, the "U/I + S" model is nested in the full three-factor model; the "C + S" model is nested in the "C + U + S" and "C + I + S" models. As shown in Table 2, the "I + U + S" model provided a better fit than the "U/I + S" model ($\chi^2_{diff} = 33.24$, $P = 6.07 \times 10^{-8}$, uncorrected), which was consistent with the previous studies[18]. For the three bifactor models, the "C + S" model did not fit as well as the "C + U + S" model ($\chi^2_{diff} = 43.29$, $P = 2.14 \times 10^{-9}$, uncorrected) or the "C + I + S" model ($\chi^2_{diff} = 29.46$, $P = 1.79 \times 10^{-6}$, uncorrected). Finally, since the "C + I + S" model and the "C + U + S" model had the same degrees of freedom (df), direct comparison of the chi-square of model-fitting suggested that the "C + U + S" model was the best bifactor model. Taken together, our results largely replicated published work.

**Correlations of the EF components with intelligence test performance.** Previous studies suggest that EFs are related to intelligence test performance[41,42]. One major question is how intelligence relates to different EF components in the unity and diversity model. Friedman et al.[43] found that common EF and updating-specific factors showed almost the same degree of correlations with intelligence test performance ($r = 0.51$ vs $r = 0.49$), whereas the shifting-specific factor showed a significant negative correlation with intelligence test performance ($r = -0.24$). We measured the participants' general intelligence with Raven's Progressive Matrices Test ($n = 924$, male = 382, age 17–31 years, mean = 20.68 years). Results showed that in the "C + U + S" model, intelligence test performance was significantly and positively associated with the common EF component ($r = 0.32$, $P < 1 \times 10^{-5}$, uncorrected) and the updating-specific component ($r = 0.14$, $P = 2.88 \times 10^{-5}$, uncorrected), but negatively associated with the shifting-specific component ($r = -0.07$, $P = 0.03$, uncorrected). These results indicated that EFs and intelligence test performance are related only to a moderate extent[4].

**Using the neural data to constrain the models based on the behavioral data**
Previous studies have examined the neural substrates of different EF components based on brain imaging data, focusing on either the "I + U + S" model[22,44] or the "C + U + S" model[28,30,31]. In this study, we used the brain image data to assess different EF models and select the best fitting one(s). The idea is that each component in a good model is expected to have dissociable and interpretable neural substrates. We chose to use resting-state functional connectivity (RSFC) patterns based on two lines of evidence. First, many cognitive functions are supported by the functional integration of distributed brain regions, which should be reflected by RSFC patterns. Consistently, previous studies have shown that RSFC patterns can successfully predict various cognitive performances[45–48]. Second, our previous study has demonstrated that RSFC patterns are significantly heritable and genetically correlated with psychiatric diseases[49]. These findings indicate that RSFC patterns can serve as optimal intermediate phenotypes[50].

**Connectome-based predictive model.** To estimate functional connectivity from resting-state fMRI data, we first parcellated the brain into 264 regions (i.e., nodes) according to the Power 264 parcellation atlas[51]. The mean BOLD time series from these nodes were used to estimate the degree of connectivity between any two nodes (i.e., edge) by calculating the Pearson correlations of their BOLD time series, resulting in 34,716 edges for each participant. We then used these edges to predict individuals' EF factor scores by applying a connectome-based predictive model (CPM)[52]. Briefly, the CPM adopts a cross-validation approach to develop predictive models of brain-behavior relationships from connectivity data. In this case, we used a linear regression with tenfold cross-validation to predict individuals' EF factor scores.

We found that when using the cutoff threshold of $P = 0.05$ to select edges, the RSFC pattern could successfully predict most EF factor scores after multiple comparisons correction using the false discovery rate (FDR) method[53] (FDR-BH corrected, $P < 0.05$) (Fig. 3), except the shifting component in the "I + U + S" ($r = 0.07$, $P_{permutation} = 0.09$, FDR-BH corrected, across 13 tests, hereafter) and the "U/I + S" model ($r = 0.06$, $P_{permutation} = 0.08$), the inhibiting-specific component in the "C + I + S" model ($r = 0.06$, $P_{permutation} = 0.09$), and the updating-specific component in the "C + U + S" model ($r = 0.04$, $P_{permutation} = 0.19$). These results suggested that the inhibiting and updating components might be mainly accounted for by the common EF component. Furthermore, we found that the RSFC could successfully predict the shifting-specific component in the bifactor models, but failed to predict the shifting component in the correlated models. This indicates that the shifting-specific component and the common component were functionally dissociated, as the only difference between the two components was whether the common component was removed from the shifting tasks. To verify the robustness of our results, we also used the cutoff of $P = 0.01$ and $P = 0.1$ to select edges and obtained remarkably similar results (Supplementary Table 4).

**Dice similarity analysis to examine the neural dissociation.** To determine whether the EF components that could be successfully predicted by the RSFC were associated with distinctive neural substrates, we examined the overlap and separation of the EF components' edges. This analysis was done on four models in which at least two components were significantly associated with brain edges.

First, we identified the edges that contributed the most to the prediction of the EF components in the CPM analysis. Edges that were selected 950 times in the 1000 iterations (100 times of ten-folds) were considered as "contributing edges"[54]. Second, we used the Dice coefficient to quantify the degree of overlap of the contributing edges for each pair of EF components in a model and tested their significance using a permutation test (see "Methods"). We found a

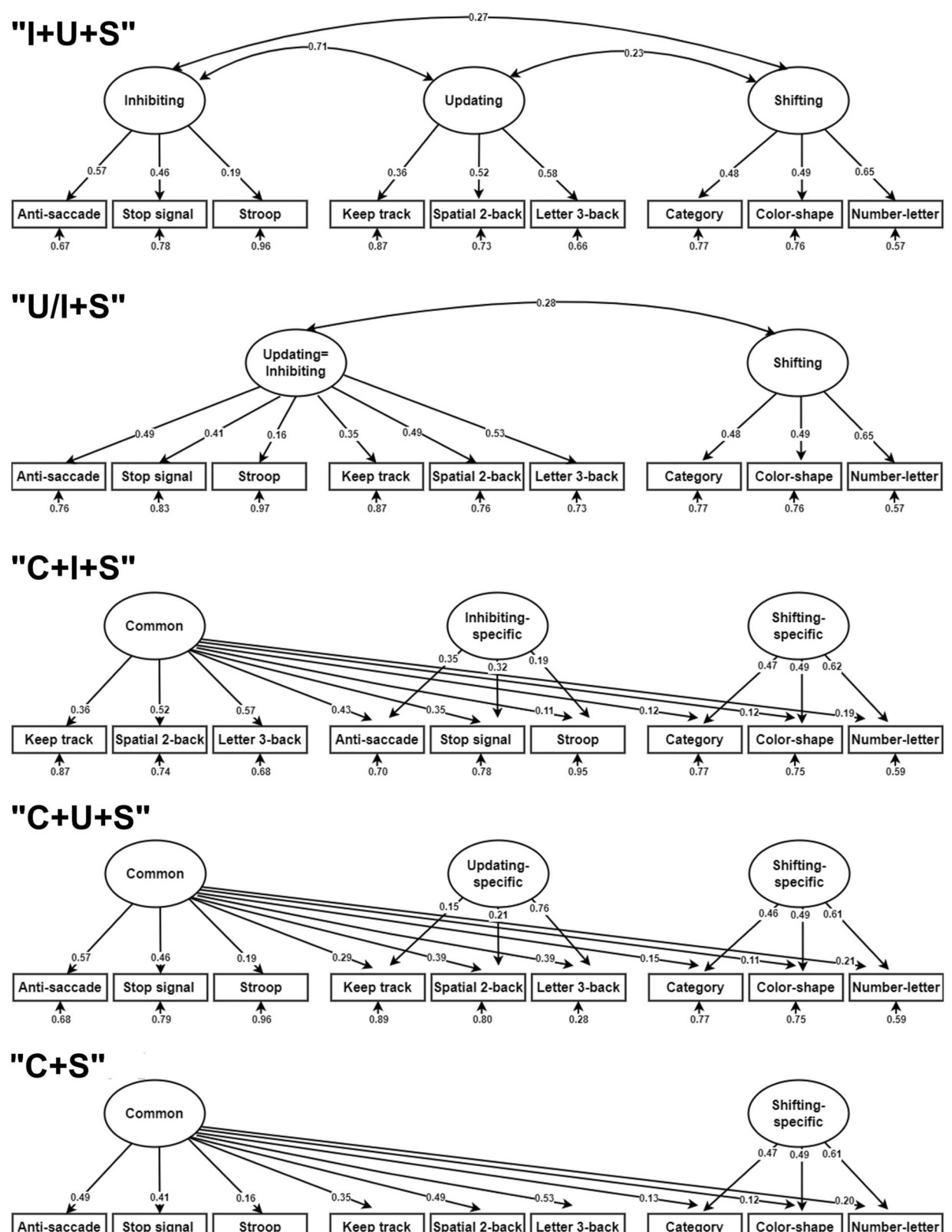

**Fig. 2 | EF latent variable models (the five good-fit models).** The numbers on the one-way arrows are standardized factor loadings between latent variables and manifest variables from the nine tasks. All the loadings had *P* values < 0.05; uncorrected; two-sided test (hereafter). The numbers on the one-way arrows at the bottom of the manifest variables are error terms. Significantly correlated pairs of latent variables are connected with double-arrowed curves and associated correlation coefficients are shown. Exact *P* values are provided in Source Data file. See Table 1 for fit statistics for the five models. Source data are provided as a Source Data file.

significant overlap of the contributing edges between the inhibiting and updating components in the "I + U + S" model (Dice coefficient = 0.58, $P_{permutation} < 1 \times 10^{-4}$, uncorrected) (Fig. 4a). In contrast, the overlap of the contributing edges between the common and shifting-specific components in the "C + I + S" (Fig. 4b), "C + U + S" (Fig. 4c) and the "C + S" bifactor model (Fig. 4d) (Dice coefficients ranged from 0.007 to 0.014, $P_{permutation} > 0.52$, uncorrected) were non-significant, suggesting dissociated neural substrates for the common and shifting-specific components.

**Table 2 | Nested model comparisons of the five good-fit models**

| Model | CFI$_{diff}$ | RMSEA$_{diff}$ | $\chi^2_{diff}$ | df$_{diff}$ | $P$ |
|---|---|---|---|---|---|
| ("I+U+S") vs ("U/I+S") | 0.03 | 0.01 | 33.24 | 2 | 6.07e-8 |
| ("C+U+S") vs ("C+S") | 0.03 | 0.02 | 43.29 | 3 | 2.14e-9 |
| ("C+I+S") vs ("C+S") | 0.02 | 0.01 | 29.46 | 3 | 1.79e-6 |

One-sided test.

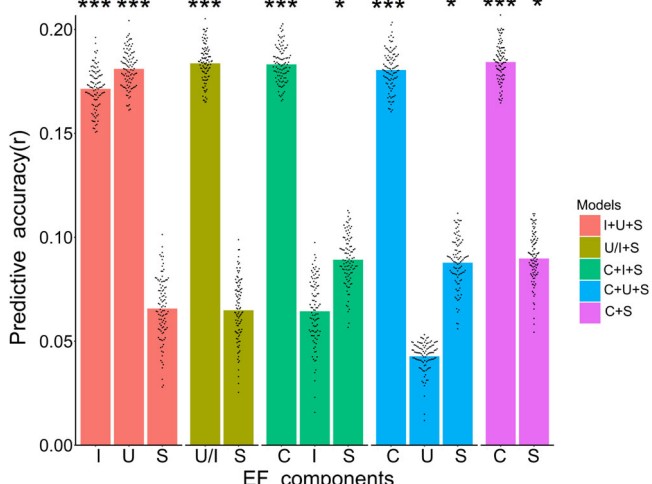

**Fig. 3 | Individual connectivity patterns predict EF factor scores across the five good-fit models.** The presented prediction accuracies (*r*) were obtained using tenfold cross-validation analyses and averaged from 100 random splits of the data (*n* = 870 subjects), each point (in total 100) overlaying the bar gragh represents the predictive accuracy of each tenfold cross-validation, *P* values were estimated using 10,000 permutations. The significant results after FDR-BH correction are noted with asterisks (\*\*\**P* values < 0.001, \**P* values < 0.05, exact *P* values are provided in Source Data file, one-sided permutation test). I inhibiting or inhibiting-specific, U updating or updating-specific, S shifting or shifting-specific, U/I (updating = inhibiting), C common. Source data are provided as a Source Data file.

Taken together, the above results suggested that the "C + S" model was best supported by the CPM results, because its shifting component was predicted with greater accuracy than was the shifting component in the "U/I + S" model, and their two components were associated with distinct edges.

## Combining CPM and Neurosynth to identify brain regions for each component in the selected model

To further validate our CPM results (based on individual differences) and to identify the brain regions associated with each component in the selected model, we conducted a conjunction analysis of our results with those from the Neurosynth meta-analysis[24] (based on group-averaged activations). Because CPM and Neurosynth meta-analysis capture different types of neural correlates of EF (i.e., correlates based on individual differences and those based on group-level analysis, respectively), the conjunction analysis provided a more robust and convergent examination of brain regions important for EF and its components.

First, we identified the specific brain regions associated with EF components in the CPM analysis. We ranked nodes based on the number of contributing edges (N) they had, and extracted the top nodes for EF components in the "C + S" model (Supplementary Table 5). For the common EF component, the top nodes were the precentral gyrus, the inferior temporal gyrus, the frontal pole, the lateral occipital cortex (LOC), the middle frontal gyrus (MFG), and the middle temporal gyrus. For the shifting-specific component, the top nodes included the LOC, the paracingulate gyrus, the planum temporale, the paracingulate gyrus, the supplementary motor cortex, the frontal orbital cortex, the postcentral gyrus, the precentral gyrus, and the central opercular cortex.

Second, we generated the Neurosynth meta-analytic maps for inhibiting (Fig. 5a), updating (Fig. 5b), and shifting (Fig. 5c) using term-based search (see "Methods", uniformity test, *z* > 3.3). A conjunction analysis of all three meta-analytic maps revealed common brain regions in the paracingulate gyrus, the superior parietal lobule extending to the superior lateral occipital cortex, the insular cortex extending to the frontal orbital cortex, and the superior parietal lobule. This conjunction map is defined as the meta-analytic results of the common EF component (Fig. 5d). We also obtained the shifting-specific meta-analytic map by subtracting the common activation regions of the three meta-analytic maps from the shifting meta-analytic map, which mainly contained the LOC, MFG, right paracingulate gyrus, and insular cortex (Fig. 5f).

Third, we did the conjunction analyses to reveal overlapping brain regions between the meta-analytic results and the CPM results. We found that for the common EF component, the strongly overlapping region was the right MFG, whereas for the shifting-specific component, the strongly overlapping regions were the right paracingulate and LOC (Fig. 5g and Supplementary Table 6).

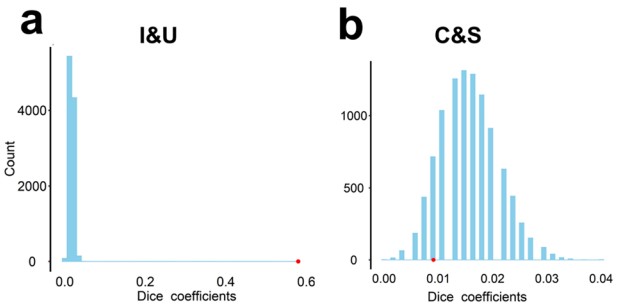

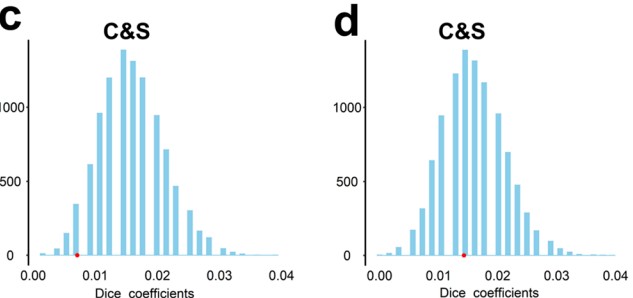

**Fig. 4 | The overlap of the contributing edges among EF components.** The bar graphs show the distribution of the Dice coefficients with 10,000 permutations. The red dot indicates the Dice coefficients obtained using the real data. **a** The overlap of the inhibiting and updating components in the "I+U+S" model. **b** The overlap of the common and shifting-specific components in the "C+I+S" model. **c** The overlap of the common and shifting-specific components in the "C+U+S" model. **d** The overlap of the common and shifting-specific components in the "C+S" model. Source data are provided as a Source Data file.

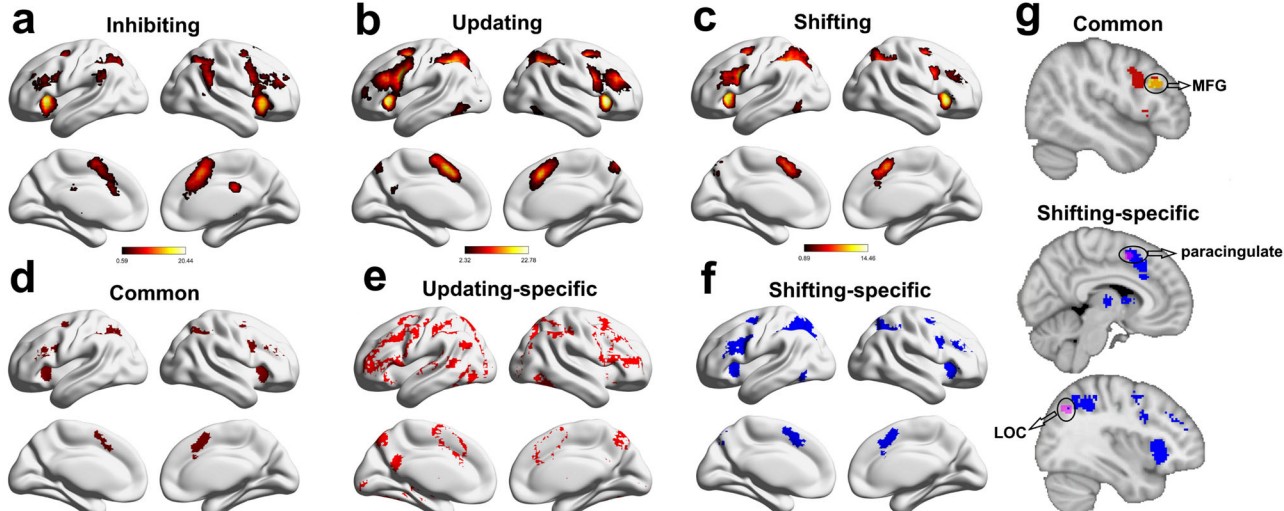

**Fig. 5 | Combining CPM and the Neurosynth results.** The Neurosynth meta-analytic results for (**a**) inhibiting, (**b**) updating, (**c**) shifting tasks, and (**d**) their conjunctions, based on the common activation map for all three tasks. **e** The updating-specific meta-analytic map was obtained by subtracting the common map from the updating-task map. **f** The shifting-specific meta-analytic map was obtained by subtracting the common map from the shifting-task map. **g** Overlap with the CPM result. The specific brain regions for the common EF component were obtained by overlapping the top ten nodes for the common EF component in the CPM analysis with the conjunction map (**d**), whereas those for the shifting-specific component were obtained by overlapping the top ten nodes for the shifting-specific component in the CPM analysis with the shifting-specific meta-analytic map (**f**). The brain figure was visualized by the BrainNet Viewer[123] under the Creative Commons Attribution (CC BY) license (https://creativecommons.org/licenses/by/4.0/). MFG middle frontal gyrus, LOC lateral occipital cortex. Here, we only show the clusters with over 100 voxels.

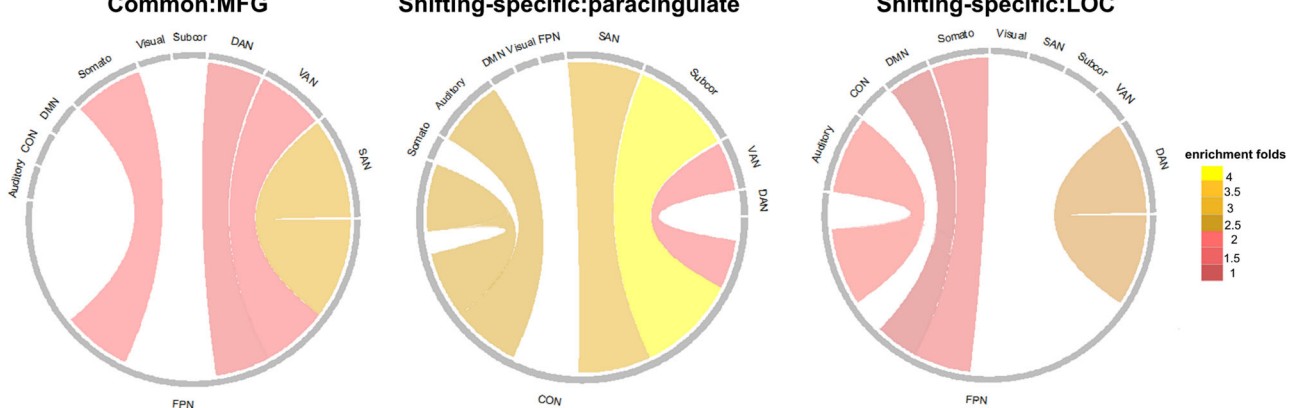

**Fig. 6 | Network enrichment patterns of contributing edges linked to key nodes for EF components of the "C + S" model.** Somato motor and somatosensory network, CON cingulo-opercular network, DMN default mode network, FPN frontoparietal network, SAN salience network, Subcor subcortical network, VAN ventral attention network, DAN dorsal attention network, MFG middle frontal gyrus, paracingulate paracingulate gyrus, LOC lateral occipital cortex. Here, we only show the network connection patterns with enrichment folds ≥1. Source data are provided as a Source Data file.

Although our CPM analysis did not reveal a significant prediction of the updating-specific component in the "C + U + S" model, previous studies have implicated the basal ganglia (BG) for the updating-specific component[4,55]. Consistently, the BG are one of the top clusters of meta-analytic results of the updating-specific component (Fig. 5e, $MNI_{(COG)} = 13, -3, 4$). Nevertheless, the edges connecting the BG could not predict the updating-specific component ($r = -0.015$, $P_{permuation} = 0.60$, uncorrected).

**The network enrichment patterns for each EF component.** Focusing on the three nodes based on the conjunction analysis mentioned above (i.e., MFG, paracingulate gyrus, LOC), we further investigated the distribution of the contributing edges that were linked to these nodes. In other words, we aimed to identify the networks in which these edges were mainly enriched[56] (see "Methods"). As shown in Fig. 6, for the common EF component, the contributing edges that were linked to the MFG (i.e., the specific brain region for the common EF component) were mainly distributed in the FPN-SAN network (2.92× enrichment). For the shifting-specific component, the contributing edges that were linked to the paracingulate gyrus were mainly distributed in the CON-Subcor network (4.05× enrichment). The contributing edges that were linked to the LOC were mainly distributed in the FPN-DAN network (2.39× enrichment).

**Using the genetic data to constrain the models based on behavioral data**
**Using genetic correlation analysis to examine genetic dissociation among EF components.** The above analyses used neural data to constrain the five good-fit models based on behavioral data, which revealed one model (i.e., the "C + S" model) that showed robust fit to

the neural data. This model showed significant and dissociated neural substrates for the common and shifting-specific EF components. Here we again started with the five good-fit models based on behavioral data and examined whether the EF components of the five good-fit models showed genetic dissociations. We expected to find such dissociations because previous studies have shown that EFs are highly heritable[4,20] and are associated with separate sets of genes[21].

First, we estimated the genome-wide SNP heritability of the EF components in the five models. We found that for the "I + U + S" model, heritability was significant for the inhibiting component ($h^2_{SNP} = 0.60$, SE = 0.23, $P = 1.03 \times 10^{-2}$, FDR-BH corrected across 13 tests, hereafter) and the updating component ($h^2_{SNP} = 0.58$, SE = 0.23, $P = 1.03 \times 10^{-3}$), but not the shifting component ($h^2_{SNP} = 0.19$, SE = 0.23, $P = 0.22$). For the "U/I + S" model, heritability was significant for the "updating = inhibiting" component ($h^2_{SNP} = 0.62$, SE = 0.23, $P = 1.03 \times 10^{-3}$), but not the shifting component ($h^2_{SNP} = 0.19$, SE = 0.23, $P = 0.22$). For the three bifactor models, i.e., C + I + S, C + U + S and C + S, heritability was moderate for the common EF component in the three models ($h^2_{SNP} = 0.58-0.63$, SE = 0.23, $P = 1.03 \times 10^{-3}$), but not significant for the shifting-specific component in the three models ($h^2_{SNP} = 0.19$, SE = 0.23, $P = 0.22$). Furthermore, heritability was not significant for the inhibiting-specific EF component in the "C + I + S" model ($h^2_{SNP} = 0.29$, SE = 0.23, $P = 0.20$) and the updating-specific EF component in the "C + U + S" model ($h^2_{SNP} = 0.14$, SE = 0.22, $P = 0.25$).

We also estimated the heritability of IQ (Raven's Progressive Matrices Test), which revealed a heritability of $h^2_{SNP} = 0.50$ (SE = 0.38, $P = 0.10$), in line with a previous GCTA analysis of 2875 children at age 12 ($h^2_{SNP} = 0.45$)[57] and a meta-analysis based on 50 years of twin studies ($h^2 = 0.54$)[58]. This value is higher than that found in a prior consortium study based on unrelated individuals[59] i.e., $h^2_{SNP} = 0.19-0.22$ across age groups; $h^2_{SNP} = 0.22$, SE = 0.10 for young adults ($n = 6033$). The higher heritability estimates may be due to the homogeneity of our sample of healthy young college students of Han ethnicity. Using the same sample, we also found higher heritability estimates of functional connectivity edges in a previous study[49] as compared to those from the UK Biobank data[60]. In addition, the limited sample size may have also led to less accurate (possibly inflated) heritability estimates.

Second, we estimated the genetic correlations between the EF components within the same models using the Bivariate GREML functions (https://yanglab.westlake.edu.cn/software/gcta/#Bivariate GREMLanalysis) in the GCTA toolbox[61,62]. Here, the genetic correlation between a pair of traits is their shared additive genetic effect, with a high genetic correlation suggesting a shared or overlapping genetic mechanism. Recall that the common and specific components in the bifactor models are orthogonal. Indeed, our results showed that the genetic correlation between the common and shifting-specific components was close to zero ($r = -0.06$, SE = 0.49, $P = 0.49$, FDR-BH corrected, across 11 tests, hereafter), which was much lower than the genetic correlations of the pairs of components in the "I + U + S" model (updating and inhibiting: $r = 0.96$, SE = 0.05, $P = 0.04$; inhibiting and shifting: $r = 0.40$, SE = 0.41, $P = 0.40$; updating and shifting: $r = 0.43$, SE = 0.43, $P = 0.40$), the "U/I + S" model ($r = 0.43$, SE = 0.40, $P = 0.40$), the "C + I + S" model (common and inhibiting-specific: $r = 0.70$, SE = 0.37, $P = 0.25$; common and shifting-specific: $r = -0.04$, SE = 0.51, $P = 0.49$; inhibiting-specific and shifting-specific: $r = -0.23$, SE = 0.73, $P = 0.49$), and the "C + U + S" model (common and updating-specific: $r = 0.61$, SE = 0.59, $P = 0.40$; common and shifting-specific: $r = -0.09$, SE = 0.50, $P = 0.49$; updating-specific and shifting-specific: $r = -0.03$, SE = 0.95, $P = 0.49$). These results suggested genetic dissociations between the common EF and shifting-specific components.

**The genetic architecture of common and shifting-specific components based on Allen Human Brain Atlas (AHBA).** To further integrate genes, brain, and cognition, we used AHBA to extract three gene sets with enhanced expression in the three brain regions associated

**Table 3 | Enrichment patterns of the Allen brain expression candidate gene sets**

| Components | Gene sets | $P_{(95\%)}$ | $P_{(75\%)}$ | EXP#genes | OBS#genes | $P_{(FDR-BH)}$ |
|---|---|---|---|---|---|---|
| Common | MFG | 0.11 | 3.0e-3 | 241 | 275 | 1.80e-2 |
| | SCG | 0.29 | 0.09 | 239 | 254 | 0.18 |
| | LOC | 0.23 | 0.26 | 242 | 249 | 0.31 |
| | BG | 0.49 | 0.08 | 240 | 256 | 0.18 |
| Updating-specific | MFG | 0.40 | 0.29 | 241 | 247 | 0.32 |
| | SCG | 0.55 | 0.19 | 239 | 249 | 0.27 |
| | LOC | 0.46 | 0.19 | 241 | 251 | 0.27 |
| | BG | 0.63 | 0.53 | 240 | 238 | 0.53 |
| Shifting-specific | MFG | 0.27 | 0.02 | 143 | 164 | 0.08 |
| | SCG | 0.22 | 2.3e-3 | 139 | 166 | 1.8e-2 |
| | LOC | 0.50 | 0.06 | 149 | 165 | 0.18 |
| | BG | 0.96 | 0.20 | 139 | 147 | 0.27 |

MFG = the set of genes that showed enhanced expression in the middle frontal gyrus; SCG = the set of genes that showed enhanced expression in the paracingulate (subcallosal cingulate) gyrus; LOC = the set of genes that showed enhanced expression in the occipital gyrus, superior division; BG = the set of genes that showed enhanced expression in the basal ganglia. $P_{(95\%)}$: nominal gene-set enrichment $P$ value for a candidate gene set (95% cutoff); $P_{(75\%)}$: nominal gene-set enrichment $P$ value for a candidate gene set (75% cutoff); EXP#genes (75%): expected number of genes that were above 75% cutoff; OBS#genes (75%): observed number of genes that were above 75% cutoff. Correction for multiple comparisons was performed with the FDR-BH. One-sided test.

with the common EF component (i.e., MFG-related genes) and shifting-specific component (i.e., paracingulate gyrus, which corresponds to subcallosal cingulate gyrus in the AHBA, SCG-related genes; and LOC-related genes, which corresponds to occipital gyrus, superior division in AHBA). For the updating-specific component, we extracted genes with enhanced expression in the BG region (see "Methods" and Supplementary Data 1 for detailed information of the candidate genes). We then estimated the enrichment pattern of the candidate gene sets for the three EF components using MAGENTA (Meta-Analysis Gene-set Enrichment of variaNT Associations) functions[63]. We used a nonparametric permutation test to examine whether there were more genes in the candidate gene set that passed the predetermined gene score rank cutoff than would be expected by chance, i.e., compared to randomly selected gene sets of identical size 10,000 times. Two enrichment cutoffs, 95 percentile and 75 percentile of all gene scores, are generally used in the literature. Given that the EFs are highly polygenic[4], we reported the 75 percentile results.

In general, our results showed that the common EF component had a different genetic basis than the shifting-specific component across different genetic boundaries (Table 3 and Supplementary Table 7). In particular, when the gene boundaries were defined as ± 35 kb, the MFG-related genes showed significant enrichment for the common EF component ($P_{75percentile} = 1.80 \times 10^{-2}$, FDR-BH corrected, across 12 tests, hereafter), but not for the shifting-specific component ($P_{75percentile} = 0.08$). By contrast, the SCG-related genes were significantly enriched for the shifting-specific component ($P_{75percentile} = 1.80 \times 10^{-2}$), but not for the common EF component ($P_{75percentile} = 0.18$). No significant results were found for the LOC-related and BG-related genes (Table 3). These two gene sets were not included in further analyses.

**Functional enrichment of the MFG- and SCG-related genes.** To further characterize the MFG- and SCG-related genes, we performed functional gene-set enrichment analysis using ToppGene suite[64] (https://toppgene.cchmc.org/). This method detects functional enrichment of a given list of genes based on gene annotations, such as Gene Ontology (GO) terms (e.g., GO molecular function, GO biological process and GO cellular component), mouse phenotypes, gene pathway, and genes associated with mental disorders. As expected, the

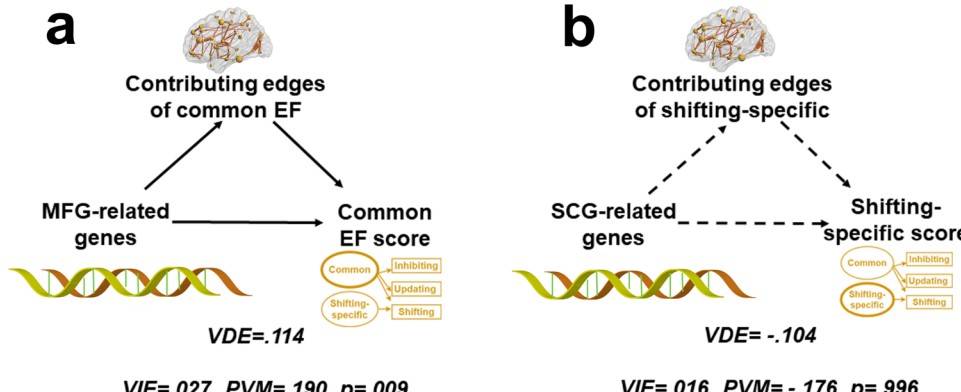

**Fig. 7 | High multidimensional mediation results. a** Mediation results of the common EF factor. **b** Mediation results of the shifting-specific factor. A significant mediation effect was found for the common EF component, but not for the shifting-specific component. The brain figure was visualized by the BrainNet Viewer[123] under the Creative Commons Attribution (CC BY) license (https://creativecommons.org/ licenses/by/4.0/). VIE variance indirect effect, VDE variance direct effect, PVM proportion of the variance mediated. The solid arrows indicate significant coefficients and dashed arrows indicate non-significant coefficients. Correction for multiple comparisons was performed with the FDR-BH. One-sided permutation test.

MFG-related and SCG-related genes showed both common and distinct enrichment patterns. In terms of GO biological process, GO cellular component, and gene pathway, both candidate gene sets showed enrichment for synaptic signaling (GO biological process, MFG-related genes: $P = 2.70 \times 10^{-53}$, FDR-BH corrected, hereafter; SCG-related genes: $P = 3.85 \times 10^{-28}$), synapse (GO cellular component, MFG-related genes: $P = 8.54 \times 10^{-52}$; SCG-related genes: $P = 7.85 \times 10^{-35}$), and neuroactive ligand-receptor interaction (gene pathway, MFG-related genes: $P = 6.49 \times 10^{-14}$; SCG-related genes: $P = 1.44 \times 10^{-5}$). In terms of GO molecular functions, MFG-related genes were enriched for gated channel activity ($P = 3.33 \times 10^{-21}$), whereas the SCG-related genes showed overrepresentation in signaling receptor binding ($P = 1.97 \times 10^{-9}$). In terms of mental disorders, MFG-related genes were enriched in schizophrenia ($P = 1.18 \times 10^{-26}$), whereas the SCG-related genes showed overrepresentation in anxiety ($P = 7.79 \times 10^{-13}$) (see Supplementary Results and Supplementary Data 1 for more results).

**The genetic architecture of common and shifting-specific components based on psychiatric or cognitive-associated gene sets.** The above analysis extracted candidate gene sets (i.e., MFG-, SCG-, LOC-, and BG-related genes) using the "gene-brain-behavior" pathway approach, which revealed dissociated gene sets for the common and shifting-specific components. To verify the genetic dissociations, we also extracted five candidate gene sets that have been associated with psychiatric or cognitive functions, such as schizophrenia-associated SNPs[65], ADHD-associated SNPs[66], intelligence-associated SNPs[59], educational attainment-associated SNPs[67], genes preferentially expressed in the central nervous system[68,69], and a negative control gene set, i.e., Crohn's disease-associated SNPs[70] (see "Methods"). Generally, our results revealed that no gene sets showed a higher contribution to the heritability of the common EF and shifting-specific components, except the intelligence-related genes and the educational attainment-associated genes. In particular, the intelligence-related genes were enriched for the common EF component when the top 30% SNPs were used (2.85× enrichment, $P = 0.02$, FDR-BH corrected, across 12 tests, hereafter)(Supplementary Fig. 3), and the educational attainment-associated SNPs showed enriched contribution to the heritability of the shifting-specific component when the top-10% SNPs were used (4.40× enrichment, $P = 4.01 \times 10^{-3}$) (Supplementary Fig. 3).

**The gene-brain-behavior pathway for common and shifting-specific components**
The above results revealed dissociated EF components in the "C + S" model at the behavioral, neural, and genetic levels. In a final analysis,

we examined whether these results converged to form a dissociated gene-brain-behavior pathway. We applied a high-dimensional mediation model[37] to investigate the relationship between the candidate gene sets based on Allen Brain expression, the contributing edges of each EF component (i.e., edges that were selected 950 times across 1000 iterations in the CPM), and the EF factor scores (see "Methods"). In this model, the high-dimensional independent measure (also called exposure in such models) was genetic variation, the high-dimensional mediator was brain functional connectivity, and the univariate dependent (or outcome) measure was the factor score of each EF component. In our analysis, we aimed to estimate the proportion of the total effect (TE) that was mediated (i.e., proportion variance mediated, PVM). A nonparametric permutation test was used to evaluate the statistical significance of the PVM, and FDR-BH was used to correct for multiple comparisons[53].

For the common EF component, our results showed that the mediation proportion was consistently significant (PVM = 0.19, $P_{\text{permutation}} = 9.4 \times 10^{-3}$, FDR-BH corrected, across six tests, hereafter) across different gene boundaries (Fig. 7 and Supplementary Table 8), suggesting that the MFG-related genes significantly affected the common EF component by regulating the functional connectivity pattern. We also investigated whether the functional connectivity edges used to predict the shifting-specific component mediated the effect of SCG-related genes on shifting-specific scores. The results indicated that the mediation proportion was not significant (PVM = −0.18, $P_{\text{permutation}} = 0.996$) (Fig. 7 and Supplementary Table 8), which was likely due to the lack of significant genetic association of shifting-specific score.

## Discussion
We investigated the structure of the EFs in a large, homogeneous, and unrelated population by integrating multimodal data from genes, brain imaging, and behavior. This approach overcomes several major limitations in existing efforts to uncover the structure of mental constructs, and provides a robust and powerful framework for ontological discovery. As an illustration of the usefulness of this approach, we found that the "C + S" model of EFs not only was parsimonious but also fit the behavioral, neural, and genetic data. These results have furthered our understanding of the structure of EFs and their neurogenetic basis, which will shed light on their theoretical development and have potential clinical implications.

We obtained convergent evidence from the behavioral, neural, and genetic data to support the "C + S" bifactor model and its "unity and diversity" structure of EF. First, by exploring the 12 candidate

models with the behavioral data, we found that the "C + S" bifactor model to have good fit. This model not only is parsimonious (as compared to the "I + U + S", "C + I + S", "C + U + S" models) but also uses two orthogonal components (as compared to the "U/I + S" model). Our results corroborate existing findings that the shifting component can be divided into common and specific components. For example, it has been reported that toddlers' self-restraint ability was not correlated with their shifting score in the "I + U + S" model at age 17, but was positively correlated with the common EF component and negatively correlated with the shifting-specific component score in the bifactor model[43]. We also found contrasting patterns in their correlations with intelligence test performance[43], which might reflect a stability-flexibility tradeoff[21,71,72].

Second, we found that at the neural level, the two EF components of the "C + S" model showed dissociated neural substrates as their contributing edges were not significantly overlapping. This is corroborated by previous reports on the dissociated neural basis of the common EF and shifting-specific components using structural MRI[28] and resting-state fMRI data[30,31]. In contrast, we did not find a significant prediction of the shifting component in the "I + U + S" or "U/I + S" model, further suggesting that the shifting component contains two distinct components (i.e., common and shifting-specific components). Furthermore, by combining the meta-analysis results from Neurosynth and the individual difference results using CPM, we found that the right MFG and the paracingulate cortex were specifically involved in the common EF and shifting-specific components, respectively. The common EF component reflects individuals' ability to actively maintain the task goal and goal-related information[21,71,73]. Correspondingly, the MFG is a core part of the multiple demand system[41,74,75]; is involved in reorienting of attention[76]; and shows activation across the shifting, inhibiting, and updating tasks[22]. In contrast, the paracingulate gyrus has been consistently associated with the shifting tasks[77–80]. Interestingly, the MFG was also consistently involved in shifting tasks[22,79], but it might be responsible for the common EF component rather than the shifting-specific component.

A few previous studies have used resting-state data and calculated simple correlations between resting-state data and behavioral performance to examine the neural correlates of common and specific aspects of EFs. For example, Reineberg et al.[30] extracted common, updating-specific, and shifting-specific components from three tasks ($n$ = 91), and found that the common EF was associated with the connectivity between the frontal pole and the attentional network, and that between the cerebellum and the right frontoparietal network. In contrast, the shifting-specific ability was associated with the connectivity between the angular gyrus and the ventral attention network. In a follow-up study using a large sample ($n$ = 250) and six EF tasks, they found that the shifting-specific component was correlated with the connectivity between the ventral attention network, particularly the cingulo-opercular subsystem, and the default mode network[31]. Using a larger sample and a cross-validation approach, we found that the contributing edges linked to the MFG for the common component were mainly located in the FPN-SAN network, whereas the contributing edges linked to the paracingulate gyrus for the shifting-specific component were mainly located in the CON-Subcor network. These results together suggest that the frontoparietal network may play a key role in common EF, whereas the cingulo-opercular subsystem may support shifting-specific EF.

Third, at the genetic level, we found that the genetic correlation between the common EF and shifting-specific components was close to zero ($r$ = −0.06), much lower than other component pairs, such as common and updating-specific components ($r$ = 0.61), and common and inhibiting-specific components ($r$ = 0.70). It is worth noting this genetic dissociation could not be simply attributed to low heritability for the shifting-specific component, as the updating-specific and inhibiting-specific components also showed low and non-significant

heritability. Yet, they showed a high genetic correlation with the common component. Furthermore, our brain expression-based gene-sets enrichment analysis showed double dissociation for the genetic basis of the two components. In particular, the MFG-related genes exhibited enrichment specifically for the common EF component, whereas the SCG-related genes showed enrichment only for the shifting-specific component.

In light of the strong correlations between the three components in the "I + U + S" model, the "C + U + S" bifactor model was developed and has been widely used in recent literature. Although the current study and previous studies found that the "C + U + S" showed the best fit to the behavioral data, our study showed that these models with a separate updating component were not fully compatible with all the cognitive, neural, and genetic data. First, consistent with previous studies, our study showed that the three components in the "I + U + S" model were highly correlated with one another at the cognitive, neural, and genetic levels and that inhibiting and updating were more highly correlated with each other than with the shifting component. Second, the loadings of the updating tasks on the updating-specific component are generally weaker than other loadings (i.e., those of the shifting tasks on the shifting-specific component). Third, the cognitive processes tapped by the updating-specific component are not yet clear. Both our study and the previous studies[20,43] found that the updating-specific component was moderately correlated with intelligence test performance beyond the common component, but it did not predict self-restraint ability[43], or procrastination[73], beyond the common EF. It has been proposed that the updating-specific component might be involved in effective gating of information and controlled retrieval from long-term memory[21]. More studies are required to examine the cognitive processes of the updating-specific component. Finally, consistent with previous studies[30,31], we found that functional connectivity patterns in the brain could not successfully predict the updating-specific factor score. Although previous studies have implicated the BG for the updating-specific component[4,55], the edges connecting the BG could not predict the updating-specific component. In addition, the genes with enhanced expression in the BG did not show any enrichment for the updating-specific component.

It is also notable that previous studies have revealed mixed results for the anatomical basis of the updating-specific component. For example, one study found that the updating-specific component was related to the dorsolateral prefrontal gray matter volume[28]. However, that study had a small sample size ($n$ = 61) and used only one task for each EF component, so its results might reflect task-specific processes rather than the latent updating-specific component. A follow-up study[81] with a larger sample size ($n$ = 251) and six EF tasks found that better updating-specific ability was associated with the greater cortical thickness of a cluster in the left cuneus/precuneus, and reduced cortical thickness in the right superior frontal gyrus and right middle/superior temporal gyrus.

Several factors could have contributed to the lack of significant neural and genetic associations with the updating-specific component. They include the relatively weak loading of updating tasks on the updating-specific component, the indeterminacy of factor scores, the reliability issue of the resting-state functional connectivity measures[82], and the missing heritability effect of the GCTA approach[83]. Future studies with significantly larger sample sizes and multimodality (structural and functional) data are needed to detect the neural and genetic associations of the updating-specific component.

Previous studies have attempted to reveal the neurobiological mechanisms of human complex behaviors through the "gene-brain-behavior" pathway[34–36]. For example, several studies have explored how the genetic effects on cognition are mediated by certain neural intermediate phenotypes[84–86]. Since these studies have focused on either one single gene or SNP, a single task, or a single brain region, the observed effects have been generally very small. Such studies are

incompatible with the fact that the EF latent components are highly heritable and highly polygenic[4]. In contrast, the current study adopted a high-dimensional mediation model[37] to examine the relationship among gene sets, brain function networks, and latent variable components of EFs.

We first used multidimensional functional connectivity patterns in the brain to predict EF factor scores. Compared with commonly used univariate traits, the multidimensional traits could provide more comprehensive information[87]. Our previous study has demonstrated that the multidimensional functional connectivity patterns are significantly heritable[49]. Many studies have also established the relationship between multidimensional functional connectivity patterns, cognitive functions, and brain disorders[45,46,48,88]. Taken together, these results confirm that multidimensional functional connectivity patterns can act as important intermediate phenotypes to link genes and behavior, thus playing a critical role in revealing the neurobiological mechanisms of human complex traits.

We then used a candidate gene-sets approach to estimate the genetic contribution to the heritability of EF components. Compared to the candidate gene approach or genome-wide association analysis (GWAS), the gene-sets approach can estimate the additive effects of multiple genes while mitigating the power issue in GWAS. In the current study, we defined gene sets based on the enhanced expression in EF component-specific brain areas, including MFG and paracingulate gyrus, according to the Allen Human Brain Atlas. The hypothesis is that such genes are likely to affect individuals' EF performance by affecting the brain's functional connectivity pattern. We also extracted candidate genes from the largest GWAS results, e.g., genes related to psychiatric disorders and cognitive abilities. Note that when we examined only the association between genes and the EF components, we found few significant results, suggesting that it is necessary to use the more holistic "gene-brain-behavior" pathway approach to identify the candidate gene sets. Hence, we finally applied a high-dimensional mediation model[37] to analyze the "gene-brain-behavior" pathway. Such a model can handle the complex, high-dimensional genetic and neural data, and capture the aggregated genetic effects on brain functions with greater statistical power[89,90]. Indeed, our model showed significant gene-brain-behavior effects.

One result from the heritability analysis is worthy of a separate comment. We found significant heritability for the common EF component, but not for the shifting-specific component. These results are consistent with previous studies. Compared to the common EF component, the shifting-specific component was found to be less heritable in a previous twin study[20]. One possible explanation is that the shifting-specific component might be more affected by environmental factors[20]. A larger sample size may be needed in order to detect the relatively small genetic effect on the shifting-specific component.

We believe that our study has important theoretical, clinical, and methodological implications. Theoretically, as one of the core cognitive functions, EFs affect cognitive processes that are important to daily life, study, and work. The "gene-brain-behavior" pathway for the common EF component revealed in the current study highlights the unity of EFs. Meanwhile, the contrasting relationship of the common EF component and the shifting-specific component with other cognitive functions emphasize the stability and flexibility framework of EFs[4]. This discovery sets the stage to further understand the relationship between EFs and other cognitive constructs, such as intelligence, attention, and memory. It could also guide future intervention studies to enhance EFs and their transfer, either by cognitive training[91–93] or targeted brain stimulation[94] or both.

Clinically, the dysfunction of EFs is considered a common risk factor across various psychiatric disorders[95,96], including schizophrenia[13], depression[97], and attention-deficit hyperactivity disorder[98]. Existing studies mainly used either complex tasks such as verbal fluency or a few simple EF tasks to estimate the

dysfunction of EFs in patients[95], making it hard to accurately quantify the nature of EF impairments related to specific aspects of psychopathology. By examining both common and specific EF components in these disease populations, we can better understand the unity and diversity across diseases and the contribution of EF dysfunctions, and consequently improve the diagnosis and treatment of psychological disorders.

Our study also provides a practical methodological framework to identify the structure of human cognition. Existing studies often try to characterize the ontology of mental constructs primarily by behavioral model-fitting, which lacks neural constraints. The large-scale meta-analytical approach provides a powerful, scalable, and relatively economical choice at the cost of precise delineation of cognitive constructs, brain activation, and brain-cognition mapping. In this study, we collected a relatively large gene-brain-behavior dataset from a healthy, homogeneous adult sample using the same behavioral paradigms, imaging and genetic data collection protocols, and analytical pipeline. Based on this rich dataset, we evaluated several candidate behavioral models and selected the optimal model based on not only behavioral data but also their consistency with multi-level evidence from genes and functional connectivity patterns in the brain.

Several methodological issues should be noted. First, although the current study recruited a relatively large sample ($n = 1454$) for the genetic analysis, which is comparable to similar previous studies[99–101], it might still be underpowered for some latent EF components such as the shifting-specific component. Second, the sample in the current study was unrelated to healthy Han Chinese young people. Although the characteristics of the sample in this study made it possible for a systematic study of the gene-brain-behavior relationships due to their genetic homogeneity, it also means that our results need to be replicated in other populations with different genetic and cultural backgrounds. In particular, the SNP sets associated with cognitive functions or psychiatric diseases were primarily based on Western participants. Although cross-population homogeneity has been reported in some studies[65,66,70], it is unknown to what extent our findings will change with SNP sets based on Asian populations, especially Han Chinese.

Third, the heritability estimates from our analysis are smaller than those based on twin studies (e.g., 0.62 for the common component in the current study vs. almost 100% in a previous twin study[20]). This "missing heritability" could be due to the fact that genome-wide SNP heritability in our study only captures additive genetic effects due to common autosomal SNPs, which ignores gene–gene[83], and gene–environment[102] interactive effects and sex chromosomes. Another factor is that genome-wide SNP heritability is also likely to suffer from low estimated heritability when causal alleles are rare[103]. A third factor is that traditional twin and family design studies may inflate the heritability estimation because of the violation of a shared common environment[104].

Future studies can be conducted along the following lines. First, future studies are needed to understand whether our findings are generalizable to other populations. Second, it would be informative to examine the structure of EFs in different age groups, because they may change with age[9,105]. Third, the three shifting tasks in the current study shared a similar task structure and behavioral index. Future studies can develop richer behavioral paradigms to examine the shifting-specific component more comprehensively. Moreover, both tasks and questionnaires can be used to improve the ecological validity of mental constructs[17]. Since cognitive measures are very sensitive to the design of tasks and their procedures, future large consortium studies need to pay particular attention to standardization of tasks and procedures in order to examine the genetic basis of cognitive ontology. Finally, it would be interesting to consider multimodality fusion technologies to integrate structural MRI, resting-state fMRI, EEG, and MEG data to improve our understanding of the neural substrates of EF components[106].

To summarize, this study provides an integrative data-driven framework to uncover the ontology of complex psychological constructs. Our framework, in combination with systematic knowledge-based approaches, such as the cognitive atlas[107,108], could push for a more detailed and systematic characterization of psychological constructs and their neural substrates and genetic basis, and advance theoretical development and translational applications in cognitive neuroscience.

## Methods

### Participants

All participants in the current study were part of the Cognitive Neurogenetic Study of Han Chinese Young Adults (CNSCYA) Project[49]. A total of 2110 unrelated Han Chinese participants (Supplementary Fig. 1, male = 845, age 17–31 years, mean = 20.65 years) with high-quality behavioral data (Supplementary Table 1, see below for task descriptions) were selected. Their behavioral data were used to estimate the covariance matrix of the nine tasks. Among these participants, 1454 participants (male = 601, age 17–31 years, mean = 20.87 years) who completed all nine tasks and had high-quality genome data were selected for further analysis. Of them, 870 (male = 349, age 17–28 years, mean = 20.96 years) also had high-quality resting-state fMRI data and were used to examine the neural mechanism of the EF latent variables. Additional subjects were recruited but excluded from further analysis due to poor task performance (40–194 subjects for the nine tasks) (see below), non-Han Chinese (9 subjects) or close genomic relationships (3 subjects) (see below), or poor-quality resting-state data (a translation greater than 3 mm in any direction, or a rotation greater than 3°, or lacking whole-brain coverage) (91 subjects)[49], or missing gender or age information (4 subjects). All participants were college students recruited from Beijing or Chongqing, China. They gave written consent to the study and were paid for their participation. This study was approved by the Institutional Review Boards (IRBs) of Beijing Normal University and Southwest University, China.

### Behavioral tasks, dependent measures, and data preprocessing

Participants were tested in a group of 30 to 40 in a computer lab, with 15–20 experimenters, each supervising two participants. The overall test lasted about 6 h, which was divided into the morning and afternoon sessions. The nine EF tasks used in this study were modified from Friedman et al.[20]. Detailed descriptions of the design and dependent measures are as follows.

**Anti-saccade.** The task was adapted from Roberts et al.[109] In each trial, a gaze point "+" was first presented in the center of the screen for a random duration between 1500 and 3500 ms, at intervals of 250 ms. Then a visual cue (a 0.32-cm black square) was presented for 150 ms on one side of the screen, followed by the target (a 0.79 cm arrow within a 1.11-cm square), which appeared on the other side of the screen for 175 ms and then shaded by a gray square. The participants had to control their attention to the target, not the cue, and pressed the left or top or right key to judge the arrow's direction. The task included 22 practice trials followed by 90 test trials. The dependent measure was the percent of error responses.

**Stop-signal.** In each trial of the task, an arrow appeared in the center of the screen with a white circle outside for 1000 ms (Go trials). Participants were asked to judge the direction of the arrow and press the left or right button quickly and accurately. In 25% of the trials (randomly selected), the white circle appeared and turned red (No-go trials), in which case participants should not respond. The delay between the red circle and the arrow was adaptive based on the participant's task performance until reaching 50% accuracy of the No-go trials. The task included four blocks, each with 64 trials. The dependent measure was stop-signal response time (SSRT). Following a previous study[110], we

first calculated the percentile Go response time of correct Go trials, based on the percentage of No-go trials that participants made response. The advantage of this method is that it does not assume that the stop rate is exactly 0.5 (an assumption that is often violated in empirical data). Finally, the SSRT was calculated as the percentile Go RT of the correct Go trials minus the mean stop-signal delay of the No-go trials.

**Color-word stroop.** The Stroop task was adapted from a classic task[111]. Specifically, we modified it by using Chinese color words, including red, green, yellow, and blue. In each trial, after a word in color appeared in the center of the screen, participants had to make a judgment about the color of the word but not the meaning of the word and press one of the four buttons quickly and correctly. In the congruent condition, each word was presented in its corresponding color (e.g., the word "green" presented in green). In the incongruent condition, the word was presented in one of the other 3 colors (e.g., the word "green" presented in red). Each word had two conditions, and each condition had 12 trials, resulting in 96 trials that were presented in a random order. The dependent measure was the response time difference between the congruent and incongruent conditions.

**Number–letter.** In each trial (160 trials in total) of the number–letter switching task[112], a number–letter pair appeared on the screen. If they appeared at the top of the screen, the participants were asked to judge if the number was odd (1,3,5,7,9) or even (2,4,6,8) and press the button as quickly and correctly as possible. If they appeared at the bottom of the screen, the participants were asked to judge if the letters are vowels (A, E, I, U) or consonants (G, K, M, R). The stimuli appeared pseudo-randomly at the top or the bottom of the screen. Trials were categorized into the repeat condition (if there was no change in judgment task from the previous trial) or switch condition (if there was task change). The dependent measure was the response time difference (the switch condition minus the repeat condition). Participants were asked to respond as quickly and accurately as possible when the stimuli appeared. The stimulus disappeared after the participants made a response. The same procedure was used for two other shifting tasks (i.e., color-shape task and category switch task, see below). For the category switch task, the stimulus disappeared after the participants made a response or after 3 s, whichever occurred first.

**Color–shape.** The color–shape switching task[113] is similar to the number–letter task but with different cues and stimuli. In each trial (160 trials in total), a cue (e.g., "YS" for color, "XZ" for shape) was presented for 150 ms on the top of screen, followed by the stimulus, i.e., a red or green circle/ triangle, in the center of the screen with the cue above it. Participants were instructed to judge the stimuli by color or shape according to the cue. The same dependent measure was collected as that in the number–letter task.

**Category switch.** The task is similar to the color–shape task. In each trial (96 trials in total) of the category switching task[114], a cue (e.g., "animacy" for living vs. nonliving judgment, "size" for size judgment) was presented by for 150 ms on the top of screen, followed by a word (stimulus) in the center of the screen with the cue above it. Participants were asked to categorize the word into (a) living or nonliving thing or (b) larger or smaller than a shoe case, according to the cue. The dependent measure was the response time difference.

**Keep track.** In each trial of the keep track task[115], 2–4 target categories of animals, countries, colors, metals, distances, and relatives appeared at the bottom of the screen, and a list of 15 words were presented in the center of the screen, one by one, each for 1500 ms. In total, 12 wordlists were tested, with 4 of which containing words of 2 categories, 4 containing 3 categories, and 4 containing 4 categories. Participants were

asked to remember the last word of each target category and write them down. The dependent measure was the total number of words that were correctly written down.

**Letter 3-back.** In the letter 3-back task[116], a sequence of 13 letters was presented in the center of the screen serially, each for 750 ms and followed by a blank screen for 2250 ms. Participants had to remember the last three letters and decide if the current letter was the same as the one shown 3 items before, and press the button within 3000 ms. Participants were given a practice session to achieve >70% accuracy or complete three practice blocks (15 trials for each block). The actual test included six blocks of 15 trials each (90 trials in total), and the dependent measure was the d-prime.

**Spatial 2-back.** The task was very similar to the letter 3-back task, but the stimuli were squares, and the task was to recall the last two squares. Ten squares were presented on the screen, with each square flashed for 500 ms sequentially with an interval of 1500 ms. Participants had to remember the last two squares and decide if the current square was the same as the one shown two items before. Participants were asked to practice for one block (24 trials). The testing phase had three blocks (72 trials in total), and the dependent measure was the d-prime.

**Data cleaning.** To ensure data quality, trials with extreme RTs and erroneous responses were excluded before averaging. Briefly, for the anti-saccade, letter 3-back, and spatial 2-back tasks, which did not use RT as dependent measures, we removed trials with an RT less than 100 ms. For other tasks, including stop-signal, color-word Stroop, category switch, color–shape switch, and number–letter switch, we first removed trials with RT less than 100 ms. We then removed the trials whose RT were 1.5× interquartile range (IQR) lower than the first quartile or 1.5× IQR higher than the third quartile. After that, we removed participants who did not make response for more than 20% of trials or made too many mistakes based on the binomial distribution. Briefly, if a participant completed a task with $N$ trials and the chance level accuracy was $r$, the least number of correct trials would be the 95% quantile of a binomial distribution B (N, r). Finally, we truncated the extreme scores (i.e., scores that were 1.5x IQR less than the first quartile or 1.5× IQR more than the third quartile) by replacing them with median −1.5× IQR and +1.5× IQR, respectively. For all tasks, we only calculated the response times of correct trials.

**Data transformation.** The cleaned dependent measures were then transformed so that a high score represented high ability. For RT indices used in the three shifting tasks, the stop-signal task, and the Stroop task, we transformed them into their corresponding negative values (e.g., -RT), the percent of error responses of the anti-saccade task and the number of correctly recalled keep track tasks were first converted to accuracy, then the arcsine transformation and logit transformation was applied, respectively.

**Regressing out the effects of age and gender.** We regressed out the effects of age and sex on the nine task indexes, and used their residuals to perform CFA. These residuals were transformed to $Z$-scores before further EF model estimation.

**Model estimation of EF latent components**
We used the "lavaan" package in R software to estimate the latent variable models with maximum likelihood estimation, and used the missing = "ML" option for CFA function. Following existing studies[117], good-fit models should meet the following criteria, comparative fit index (CFI) > 0.95, standardized root-mean-square residual (SRMR) < 0.05, and root-mean square error of approximation (RMSEA) < 0.05. We did not use chi-square ($\chi^2$) as it is almost always statistically significant for models with more than 400 cases and is also affected by

the degree of the correlations in the model (the larger the correlations, the poorer the fit). The index $\chi^2$/df is also problematic since there is no universal agreement on what is a good (or bad) model.

**Resting-state fMRI data preprocessing and network construction**
We collected neuroimaging data using 3.0 T Siemens MRI Trio scanners in the Brain Imaging Centers at Beijing Normal University and Southwest University. During the resting-state scan, participants laid supine on the scanner bed and closed their eyes, and they were asked not to think about anything special. A gradient echo EPI sequence with PACE was used for functional scanning. For the Beijing sample, we used the following parameters: TR = 2000 ms; TE = 30 ms; flip angle = 90°; FOV = 200 × 200 mm$^2$; 64 × 64 matrix size with a resolution of 3.1 × 3.1 mm$^2$; thirty-three 3.5 mm transverse slices. A total of 200 brain volumes (time points) were acquired. For the Chongqing sample, FOV = 200 × 200 mm$^2$; a 3.4 × 3.4 mm$^2$ inplane resolution, and thirty-two 3.0-mm transverse slices were used to acquire a total of 242 volumes. We used GRETNA[118] tools and the AFNI[119] software to preprocess the resting-state fMRI data according to standard steps, including deleting the first 10 EPI volumes, slice-timing correction, realigning, normalization, adjusting for the nuisance covariates, and removing linear trends using temporal filters (0.01–0.1 HZ) in a single regression model[120]. In particular, we included the global signal, the average signal of the white matter and the cerebrospinal fluid, and the 24 motion parameters as nuisance covariates[121]. Then we used the Power 264 parcellation scheme[51] to assess functional connectivity across the 264 nodes of the whole brain, resulting in 34716 edges. To control for the differences in scanning parameters, the connectivity strength of all edges was first normalized within each participant before conducting the CPM. More detailed information about the imaging data acquisition, preprocessing, and functional network construction can be found in our previous study[49].

**Linking EF latent components with functional connectivity patterns**
We applied a modified CPM protocol[52] to explore the neural mechanism of each EF component. The CPM is a data-driven approach to developing predictive models of brain-behavior relationships from connectivity data using cross-validation, which includes four main steps: (1) feature (i.e., brain edges) selection, (2) feature summarization, (3) model building and application, and (4) estimation of prediction significance. Using CPM, we can get a generalizable model, which uses brain connectivity data as input and generates predictions of behavioral data in novel subjects.

Following Rapuano et al.[122], to avoid data contamination between the behavioral models and the neuroimaging analysis, component scores were recomputed using the behavioral subset of participants ($n = 1240$). The resulting component scores were consistent with those obtained when the whole sample was used ($r > 0.9$). The loadings from the behavioral subset of participants were subsequently used to transform EF performance data in the CPM analysis ($n = 870$). Furthermore, to avoid biasing the test set, edge strengths were standardized (z-scores) across subjects within each fold of the training set and the test set, separately.

We used tenfold instead of leave-one-out cross-validation in consideration of our large sample size ($n = 870$). The 870 participants were randomly divided into ten groups. We did 100 random splits of the data, and the results were averaged. Each time, we left one group of participants out as the testing group and used the remaining participants to build the training model. We calculated partial Pearson correlation between each edge (34,716 in total) and each EF factor score, controlling for the effect of fMRI scanner, and head motion (mean framewise displacement, FD). The most relevant edges ($P \le 0.05$) were selected and used to predict the testing participants' EF factor scores.

We calculated four unit-weighted summary scores of the connectivity strength values: all selected positive edges and all selected negative edges for the training and testing sets separately. To build the predictive model, we used the summary scores of the positive and negative edges from the training set to predict the EF factor score in a linear regression. This predictive model was then used to predict the EF factor score in the testing set. The Pearson's correlation coefficient between the predicted EF factor scores and their true EF factor scores was used to index the accuracy of the predictive model. The statistical significance of the predictive accuracy was estimated using 10,000 permutations, in each of which we shuffled EF factor scores and repeated the above analysis, and computed a correlation coefficient. The 10,000 correlation coefficients were used to construct the null distribution. Multiple comparisons correction was performed using FDR[53]. To ensure that our results were not critically determined by the threshold we used to select edges, we also repeated the above analyses using several different thresholds, such as $P = 0.01$, $P = 0.1$.

### Dice similarity analysis to examine the independence of neural substrates

To examine the neural dissociations between different EF components in the same model, we first identified edges that were selected 950 times across the 1000 iterations in the CPM model ($P \leq 0.05$) and defined these edges as "contributing edges". Then, we calculated the degree of overlap of the contributing edges between any pair of components of the same model. We used the "Dice similarity coefficient" to quantify the degree of overlap of the contributing edges, which is defined as:

$$Dice(X,Y) = \frac{2|X \cap Y|}{|X| + |Y|}$$

where X, Y are two sets, |X| and |Y| denote the number of elements in set X and Y, respectively; $\cap$ represents the intersection operation to obtain the same elements of two sets. We examined the statistical significance of a Dice coefficient using 10,000 resampled data sets, in each of which we randomly selected |X| and |Y| edges for the first and second components, respectively, and used them to calculate the Dice similarity coefficient. Note that in each resamapled dataset, each edge was either selected or unselected for each component. Thus, this procedure is equivalent to permutating the edge selection status 10,000 times for each component. The resulting 10,000 Dice coefficients were used to construct the null distribution. Multiple comparisons correction was performed using FDR[53].

### Comparing the neural results with the meta-analytic map

To compare our results with those of the previous studies, we first selected contributing edges in the CPM model ($P \leq 0.05$). We then ranked the nodes based on the number of contributing edges ($N$) connecting them.

Second, we extracted the fMRI meta-analytic results using term-based meta-analyses in the Neurosynth[24] (https://www.neurosynth.org/) and the meta-analytic results were visualized with the BrainNet Viewer (v1.62, http://www.nitrc.org/projects/bnv/)[123]. In brief, studies that mentioned a specific term (e.g., inhibiting) at least once in their abstracts were included in the meta-analysis. We downloaded the uniformity test map, which displays consistently active brain regions (after FDR correction, $P < 0.01$) across studies. The z-scores of these regions were calculated using a chi-square ($\chi^2$) test, where the null hypothesis is that activation in all the brain regions was equally likely. Thus, voxels with large z-scores are reported more often in studies than the other voxels. In particular, we used term-based search to extract related meta-analytic maps (uniformity test, $z > 3.3$). The "inhibiting" meta-analytic map was obtained by averaging the maps from search terms "inhibit", "inhibition", "inhibitory", "inhibitory control", "response inhibition", and "stop". Similarly, the maps from terms "updating" and "working memory" were averaged, resulting in "updating" meta-analytic map. No map was generated using the search term "update". For the "shifting" meta-analytic map, it was obtained by averaging the maps of search term "shifting", "shifts", "switching", and "switch".

Third, we picked out the top ten nodes of the EF components based on the CPM analysis and performed a conjunction analysis with the corresponding meta-analytic maps. The resulting brain regions can be seen as the specific brain regions for certain EF components.

Focusing on these specific brain regions (nodes), we estimated the network enrichment patterns of the contributing edges that were connected to these nodes. The enrichment fold was computed as the ratio of the actual observed number of selected edges within the network ($A_l$) and the expected number of selected edges ($E_l$). $E_l$ was calculated as the number of total edges that can connect to the certain node within the network multiplied by the ratio of the number of selected edges connecting to the certain node and number of all edges that may connect to the node (i.e., 263).

### Genetic data preprocessing and imputation

The detailed genotype quality control and imputation process were presented in our previous study[49]. The study samples were genotyped using one of three Illumina chips, including Illumina OmniExpress, Illumina Zhonghua, and Illumina Omni2.5. We used Plink 1.9[124] (https://www.cog-genomics.org/plink2) to perform standard genome-wide association quality control filters. In detail, single nucleotide polymorphisms (SNPs) were excluded if they had minor allele frequency (MAF) of <5%, or per-SNP missingness >5%, or a failing of the Hardy–Weinberg equilibrium test ($P < 1 \times 10^{-6}$). We also excluded participants with missing SNPs >5%. Finally, 52,6101 autosome SNPs met QC for Illumina OmniExpress chip, 671,348 autosome SNPs met QC for Illumina Zhonghua chip, and 514,369 autosome SNPs met QC for Illumina Omini2.5. These cleaned SNPs were imputed against the 1000 Genomes reference panel (see details in our previous study[49]). SNPs with imputation information score $R^2 \geq 0.3$, Hardy–Weinberg $P$ value $\geq 1 \times 10^{-6}$, $MAF \geq 1\%$, and per-SNP missingness $\leq 5\%$ were kept. In total, ~6.8 million common autosomal chromosome SNPs were included in further analyses. To remove close relatives, we first estimated the genetic relationship for each pair of participants using GCTA functions[62,125] (https://yanglab.westlake.edu.cn/software/gcta/#MakingaGRMresulting), which results in a genetic relationship matrix (GRM). Each value in the GRM reflects the average correlation of SNP values between two participants over a number of SNPs. Three pairs of participants showed estimated genetic relatedness >0.05, so one participant from each pair was randomly selected and removed. In addition, to check the ancestry of the participants (e.g., Han Chinese), we performed the principal component analysis (PCA)[126] implemented in the GCTA software[62]. Nine participants were identified as non-Han Chinese and were then removed, leaving a final sample of 2110 Han Chinese participants for further analyses (Supplementary Fig. 1).

### Estimating the genetic correlations of the EF components

We applied the GCTA functions[62] (version 1.26, https://cnsgenomics.com/software/gcta/) to estimate genome-wide heritability and the genetic correlations of EF components using the whole-genome data. Briefly, we first estimated the pairwise genetic relationship matrix (GRM) in a large-scale unrelated population by using dense SNPs. Then, we fitted the GRM in a linear mixed-effects model to estimate the heritability of a trait. For the genetic correlation analyses, we fit the GRM in a bivariate linear mixed-effect model[127] to estimate the genetic correlation of two traits[128]. To avoid spurious association due to subpopulation stratification, the principal component analysis (PCA) was performed using functions in GCTA[62] and the top ten ancestral

principal components (PCs) were used. In addition, age, gender, genotype array, and site (i.e., Beijing or Chongqing) were also included as covariates in the model. We used the FDR method for multiple comparisons correction[53].

## Defining the candidate gene sets

To explore the genetic mechanism of each EF component, we first extracted several candidate gene sets based on gene expression or previous GWAS results. We extracted genes that showed enhanced expression in EF component-specific brain regions, as compared to the whole brain, using the "Allen Brain Atlas" (https://portal.brain-map.org/) human microarray dataset and the "differential search function"[129,130]. In particular, we used a preprocessing pipeline[131] to preprocess the genes. Briefly, we first reannotated probes to genes with information from ref. 131. We filtered probes that did not exceed the background noise in more than 50% samples, which resulted in 31,569 probes (15,633 genes) for each tissue sample. These preprocessing steps were done using abagen (https://github.com/netneurolab/abagen). We sorted genes by fold change (log ratio of expression) and kept the top 1000 genes as candidate gene sets. Finally, we extracted the SNPs within 35 kb (or 25 kb, 50 kb) upstream and downstream from the 3′ and 5′ untranslated regions of each gene according to the UCSC hg19 assembly.

We also extracted five gene sets based on previous studies: genes preferentially expressed in the CNS[68,69], SNPs associated with human intelligence test performance[59] (https://ctg.cncr.nl/), SNPs associated with educational attainment[67] (https://www.thessgac.org/data), SCZ-associated SNPs[65] (https://www.med.unc.edu/pgc/results-anddownloads/), ADHD-associated SNPs[66] (https://www.med.unc.edu/pgc/results-anddownloads/), and Crohn's disease-associated SNPs[70] (https://www.ibdgenetics.org/#downloads), the last of which was used as a negative control gene set. For the functionally linked genes and central nervous network genes, we defined genic boundaries as 50 kb upstream and downstream from the 3′ and 5′ untranslated regions (UTRs) of each gene according to UCSC hg19 assembly. For the rest of the gene sets, we ranked the imputed SNPs in our study based on $P$ values from the GWAS summary statistic, and selected the relevant SNPs (e.g., top 10%, top 20%, top 30%) as associated SNPs of the corresponding traits.

## Gene-set enrichment analysis using MAGENTA

We used functions in MAGENTA to estimate the enrichment pattern of the candidate gene sets related to the EF components. It tests whether candidate gene sets are enriched for genes associated with a given complex trait, more than would be expected by chance. We used the GWAS results as input and applied the default settings. Briefly, we first mapped SNPs and their association scores ($P$ values) onto genes; then we scored each gene using the most significant SNP $P$ value; third, we corrected for confounding effects on the gene scores (e.g., gene sizes); finally, we calculated a gene-set enrichment $P$ value for each candidate gene set. Multiple comparisons were corrected using FDR[53].

## Partitioning heritability enrichment analysis

The genome data was divided into two sets: one was the trait-associated SNPs, and the other was defined as control SNPs. We calculated two GRMs and used a joint analysis[132] to estimate the genome-wide SNP heritability as the sum of $h^2_{set}$ (heritability attributed to candidate gene sets) and $h^2_{Control}$ (heritability attributed to the unselected SNPs). Then, we calculated enrichment folds for each gene set as the ratio of the estimated $h^2_{set}$ to the expected $h^2_{set\ (expect)}$, using the following formula:

$$\text{enrichment}(X) = \left( \frac{h2\text{set}}{h2\text{set(expect)}} \right)$$

Here, the expected $h^2_{set\ (expect)}$ was the genome-wide heritability $h^2_g$ multiplied by the percentage of the SNPs in the given set among all SNPs. We then computed the $z$-score to determine the significance level[49,133]. The same covariates as in the genome-wide heritability analyses were used. The FDR method was used for multiple comparisons correction[53].

## High-dimensional mediation analysis

We used the MedMix method to perform high-dimensional mediation analyses ($n = 870$) of "gene set−functional connectivity−EF factor scores". This method[37] was designed for mediation analysis of high-dimensional independent measurements, high-dimensional mediators, and a univariate dependent outcome. In brief, the dependent measure Y is an n × 1 vector, the independent measure Z is an n × q matrix, the mediator M is an n × p matrix, and Y, M, and Z are centered by column. See the following equations:

$$Y = M\gamma + Z\beta + \epsilon$$
$$M_j = ZB_j + \eta_j \, \mathrm{j} = 1, \ldots, \mathrm{p} \tag{1}$$

In the genome, most gene effect sizes might be weak, but not equal to zero. Therefore, just like the SNP-based heritability studies, here the effects of Z were modeled as random effects to reduce the dimension of the parameter space. In other words, both $\beta$ and Bj are assumed to follow multivariate normal distributions. This practice is quite common in genetic studies, as the effect sizes of individual genes are typically small. Given that functional edges are also high-dimensional, it is reasonable to focus on the proportion of total genetic effect on the variance of EFs that can be mediated by brain functional connection edges. Thus, the quantity of interest is the proportion of the variance mediated (PVM), which is defined as the ratio of the variance indirect effect (VIE) to the variance total effect (VTE):

$$VTE = (B\gamma + \beta)^T Var(z)(B\gamma + \beta), VIE = (B\gamma)^T Var(z)(B\gamma)$$
$$PVM = VIE/VTE \tag{2}$$

Where Z is a length q vector, and B is a $q \times p$ matrix whose $j$th column is $B_j$.

## Reporting summary

Further information on research design is available in the Nature Research Reporting Summary linked to this article.

## Data availability

The AHBA is available at https://human.brain-map.org/static/download/. The Neurosynth database is available at https://neurosynth.org/. The sources of the GWAS summary results are as follows: schizophrenia and ADHD (https://www.med.unc.edu/pgc/results-anddownloads/); intelligence test performance (https://ctg.cncr.nl/); educational attainment (https://www.thessgac.org/data); and Crohn's disease (https://www.ibdgenetics.org/#downloads). The list of genes preferentially expressed in the central nervous system was obtained from the corresponding author of the study[68]. The UCSC hg19 assembly: UCSC Human Gene Sorter. Behavioral data to estimate EFs models can be found in Supplementary Data 1. Raw data of the Cognitive Neurogenetic Study of Han Chinese Young Adults (CNSCYA) Project are available from the corresponding author on reasonable request. Restriction of raw data is to protect the privacy of participants. Source data are provided with this paper.

## Code availability

Custom codes are variable at https://github.com/psychelzh/Struct_EF.

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

## Acknowledgements

This work was sponsored by the National Natural Science Foundation of China (31730038 to G.X.), 973 Program (2014CB846102 to G.X.), the 111 Project (B07008 to G.X.), the NSFC and the Israel Science Foundation (ISF) joint project (31861143040 to G.X.), the Sino-German Collaborative Research Project "Cross-modal Learning" (NSFC 62061136001/DFG TRR169 to G.X.), and the Guangdong Pearl River Talents Plan Innovative and Entrepreneurial Team grant #2016ZT06S220 to G.X.

## Author contributions

G.X., Q.D., and Chuansheng C. conceived the project, designed its components. J.F. and G.X. carried out data analysis, and discussed the interpretation of data. L.Z. helped with EF latent variable models, Chunhui C. helped with the genome data imputation and preprocessing, J.S. helped with the resting-state data preprocessing. Zhaoxia. Y. wrote the scripts for multidimensional mediation analysis. Chunhui C., B.Z., J.F., L.Z., Zhifang. Y., K.F., J.L., and C.Y. participated in data collection. J.F., G.X., Chuansheng C., and Zhaoxia. Y. prepared the manuscript, and all authors critically reviewed and approved the manuscript.

## Competing interests

The authors declare no competing interests.
