## [Peer Review File · Nature Communications]

A cognitive neurogenetic approach to uncovering the structure of executive functionsEditorial Note: This manuscript has been previously reviewed at another journal that is not operating a transparent peer review scheme. The manuscript was considered suitable for publication without further review at *Nature Communications*.

REVIEWER COMMENTS

Reviewer #1 (Remarks to the Author):

I want to reiterate that this paper includes a lot of really interesting analyses and adds to the field. Additionally, the current version is an improvement on the previous submission. A number of the edits that the authors made significantly improved the manuscript.

There are still areas for improvement. The main remaining issues surround overstatement of some conclusions (which can be reframed and/or softened), precision of language for clarity, and further motivation for certain analytic choices.

1) Some of the conclusions drawn by authors still seem a bit overstated.

For example in the section discussing the genetic correlations it is stated, "These results demonstrated that the common EF and shifting-specific components showed dissociable patterns at the genetic level." Yes, their correlation between the common EF and shifting-spec scores was close to zero (as expected - like they point out) however, only one correlation out of all of the others that they ran was significant after correction for multiple testing. While the correlation coefficients were larger for a number of the other pairs, given the lack of significant findings (likely due to imprecision around the estimates) make it hard to draw the conclusion that the lack of correlation between cEF and Shift-spec is notable.

Additionally, the conclusion that the cEF+shi-spe model was the best fitting model overall is too strongly stated. It was not the best fitting behavioral model. There was some evidence for it being a compelling neural model, which then they carried forward as the only model for some of their subsequent analyses. For any of those analyses, it is misleading to say it was the best fitting model, because they did not compare it to others. Additionally, when they did compare it to other models, it may have performed better for technical reasons that have been brought up in prior comments. The authors state that the models including a separate component for updating are "not fully compatible with all the cognitive, neural, and genetic data." However it was consistent with the cognitive data and was rarely tested in the genetic data. For example the evidence for genetic influence on shifting was as weak, if not weaker than updating in the gcta heritability analyses. The only other time they look at updating in relation to any of their genetic analyses is with the genetic correlations, and I discuss that previously.

If the authors were to reframe this section to focus on the fact that there were converging lines of evidence that suggest that the cEF and shifting-specific components may be neurally and genetically distinct, that would be well supported by these sets of analyses. However the focus on "best model" seems somewhat distracting from the most interesting parts of this paper.

2) Precision in language - at times when the authors are comparing the different models, it's hard to tell if they're referring to updating or shifting "components" from the correlated model or if they really mean updating-specific and shifting-specific cognitive processing from the bifactor models. Therefore, it would be useful to consistently refer to the updating- and shifting-specific abilities as such, and then continue to refer to the updating, inhibiting, or shifting abilities as they currently do.

Phenotypic Factor Model results:

Best fitting model: Common EF + Updating-spec + Shifting-spec

Neural Data:

Best fitting model: Common EF + Shifting-specific

Neurosynth Section:

The authors clarify that their neural data is based-on individual differences and the neurosynth maps are based on group averages. However, it is not directly addressed that areas important for individual differences in performance do not necessarily map onto the regions that are used in general (or on average) to complete a task. While the overlapping regions show some sort of convergence that those areas are important for those abilities (and possibly adds validity to the CPM methods), it would be useful to point out that

Node Enrichment Section:

It is unclear to me why this was done instead of conducting a more typical resting-state network analysis. It could be because this has already been done in other samples, however it would be useful to address what this tells us above and beyond those results.

AHBA section:

Did you test other thresholds other than $\pm 50\text{kb}$? $\pm 35\text{kb}$ has recently been frequently cited in the literature (Wu, Y., Zheng, Z., Visscher, P. M. & Yang, J. Quantifying the mapping precision of genome-wide association studies using whole-genome sequencing data. *Genome Biol* 18, (2017))

In a later section you state that you used: 50kb (or 25kb, 75kb,100kb), however it seems more pertinent that you use smaller windows rather than larger (e.g., Palmer RHC, Benca-Bachman CE, Huggett SB, et al. Multi-omic and multi-species meta-analyses of nicotine consumption. *Transl Psychiatry*. 2021;11(1):98. Published 2021 Feb 4. doi:10.1038/s41398-021-01231-y)

The genetic architecture of common and shifting-specific components based on psychiatric or cognitive associated gene sets:

Please motivate the use of this enrichment method (rather than using polygenic scores to predict the cEF and Shifting-specific scores). What does this analysis tell us that a PGS would does not?

Reviewer #2 (Remarks to the Author):

This is a revision of a manuscript for which I was a reviewer when it was submitted to a different Nature Journal. Overall, I found the authors very responsive to the prior reviews. However, I do still have some concerns about the primary conclusion and some remaining questions about details. I detail these concerns and questions in the following paragraphs, but the bottom line is that overall, I think this is an impressive study that will be of great interest to the field, and I believe my concerns could be addressed with another revision.

Most importantly, I am glad that the authors have added the discussion of the weak updating-specific results and limitations in the discussion, but I remain uncomfortable with some of the statements in the text. In particular, the authors continue to characterize the C+S model as the “best-fit model” (line 682). I continue to believe it is misleading to characterize this as the “best fit” model for several reasons:

First, it is statistically worse than models with an updating-specific component in terms of chi-square difference tests, and shows large enough change in CFI/RMSEA to warrant the conclusion that it is providing poorer fit than the C+U+S model (which is also supported in the prior literature) or the C + I + S model. For example, studies examining model comparisons in the context of invariance (e.g., <https://doi.org/10.1080/10705510701301834>) have used a criterion of change in CFI > .01 and change in RMSEA > .015.

Second, some of the comparisons the authors make based on the neural and genetic data are inconsistent. For example, on line 381: “Taken together, the above results suggested that the “C+S” model was the best-fit model based on the neural data, because its shifting component was predicted

with greater accuracy than was the shifting component in the “U/I+S” model, and because it had a greater degree of dissociated edges for each component than did the “U/I+S” model or the “I+U+S” model. These results indicate that there are specific neural substrates for the shifting component, but the neural bases of the inhibiting and updating components are likely to be highly overlapping.” I don’t disagree that the neural components for U and I are likely to be overlapping, but this conclusion that the C+S model is best seems unwarranted given that the comparisons the authors are making were not tested for significant differences and ignore some non-differences (the shifting component seems equally well predicted in the C+I+S, C+U+S, and C+S models, so why only focus on comparing the C+S to the correlated factors models?). Moreover, by this logic, the C+U+S or C+I+S models should be preferred because they show better prediction of C than the C+S model.

With respect to the genetic data, these genetic analyses are very interesting, but the authors don’t really comment until late in the discussion on the low, non-significant heritability of the shifting component across models. As the authors note, it may be just a power issue, given that twin studies have found lower heritability for shifting components, but it does beg the question of how these results are constraining model selection, given that in this sample there is little evidence for unique genetic variance in shifting, at least at the genome-wide level. Of course, it makes sense that the authors don’t put too much weight on this point given that this sample size has very low power for these kinds of analyses, yet it does present an inconsistency in terms of how they are using the results to constrain their model selection.

Third, and more generally, I worry about the claim that this limited set of neural data can constrain model selection. Perhaps it can, but my concern is that the authors have only examined resting state data. Although I think it would be a lot to try to add structural measures to the analyses, it would be worth discussing whether other neural measures may have resulted in different conclusions. The authors justify their use of RSFC as “optimal intermediate phenotypes” (line 313), but in prior studies resting state does not always predict outcomes strongly and generally shows lower reliability and heritability compared to structural brain measures, as illustrated by some of the citations I listed in my prior review. In fact, prior studies examining brain predictors of the unity/diversity model specifically (which were cited by the authors here) did not find RSFC measures that predicted the updating-specific component (Reineberg et al., 2015), but *did* find *structural* correlates of updating-specific (Smolker et al., 2015; 2018). Thus, it could be that the conclusion that the updating-specific component is only weakly discriminated is specific to resting state data.

Some of the discussion is selective about citing supporting evidence for dissociating shifting, ignoring evidence that might speak to the validity of the updating-specific component. For example, Line 748: “Fourth, a previous study found that the self-restraint ability was significantly associated with the common EF ability at age 17, but not with the updating-specific ability⁴¹, suggesting that the updating-specific might not make unique contributions to the self-restraint beyond the common component.” I don’t follow why the authors are focusing on this one study here. Friedman and Miyake (2017) discussed how different outcomes show different profiles of relationships with the EF components, such

that updating-specific is generally not related to psychopathology or self-restraint, but is related to intelligence, whereas psychopathology and self-restraint is most closely related to the common component, with only small negative relationships to the shifting-specific component for some behaviors. So by this logic, the validity of the updating-specific component is strong given this prior work suggesting discriminant validity.

Moreover, many of the follow-up models (e.g., using neurosynth, gene sets, etc.) focus on the C and S components in the C+S model, but some of the same analyses could be extended to include the U component in the bifactor models. I would be interested to see what neural and genetic areas may be suggested by the neurosynth maps and gene pathways, if any.

All this said, I do still think this is an impressive study and set of analyses, and I am glad to see how consistent the results are in this sample vs. others at a general level. I am not necessarily suggesting that the authors add substantial analyses (such as adding structural data), although examining some of the neurosynth and genetic results for the updating specific factor would be appreciated. I believe that the authors could address my primary concerns by removing remaining references to “best fit” models. Perhaps rather than framing the goal as selecting the best EF model, the authors could frame this more as a characterization of the unity and diversity components in terms of RSFC and genomic variance, with the conclusion about the updating-specific component being that it is relatively weaker, at least in terms of the sample and metrics examined here.

Specific comments:

Results:

Given that both nested and non-nested models are compared, it would be useful to include information criteria (AIC and BIC) in addition to the CFI, RMSEA, and SRMR. In addition, nested models can be compared with chi-square difference tests and change in CFI/RMSEA.

I was also quite surprised at the high h^2_{SNP} , as I believe those are much stronger than typical reports for cognitive measures. Have the authors calculated heritability of their IQ measure to see if it is also higher than prior reports?

Discussion:

Line 723: “It is worth noting this genetic dissociation should not be purely contributed to low heritability for the shift-specific component, as the updating-specific and shift-specific components also showed low and insignificant heritability.” Perhaps the authors meant to say “the updating -specific component also showed low and insignificant heritability”?

Method:

Thank you for adding more information about the methods, but I did still have some questions:

Please clarify the total number of trials for the Stroop task, the switch tasks, and the keep track task.

What was the response to cue interval for the switch tasks?

The “letter memory” task is really a 3-back task; it should not be called letter memory to avoid confusion with prior studies.

Line 985: “The stimulus disappeared until participants made a response.” I think the authors mean that the stimulus disappeared after the participants made a response?

Please clarify how internal consistency was computed (e.g., one could use odd/even trials, correlations between blocks, etc., and were correlations corrected with the Spearman-Brown prophecy formula)? It’s also not clear why consistency could not be computed for stop-signal and keep track; for stop-signal the SSRT can be calculated for each block and a Cronbach’s alpha estimated; for keep track prior studies have used a Cronbach’s alpha computed on subsets of trials (e.g., the average of the first trials at each difficulty level, the average of the second trials at each difficulty level, and so on).

Line 1109: “To build the predictive model, we summarized, for each participant in the training set, the selected edges’ strength values to a single connectivity value and used it to predict the EF factor score with a linear regression.” Please clarify what “summarized” means. One could add the strength values with unit weighting, do a weighted sum; moreover, how were positive vs. negative predictions treated?

Reviewer #3 (Remarks to the Author):

The paper by Feng and co authors, titled “A cognitive neurogenetic approach to uncovering the structure of 2 executive functions”, and submitted to the journal Nature Communications, presents a three fold study of behavior, neuroimaging, and genetics to understand the neurogenetics underlying of executive function. Giving the wide area of study and the previous reviews/reviewers expertise, I focused my review on the neuroimaging aspect of this work. Overall, while there is an impressive

amount of effort that went into this study, I find the neuroimaging aspect to have a few methodological flaws, which limit my enthusiasm for the study.

There is a data contamination problem with the cross validation strategy used. The authors first construct latent factors of executive function based on the total sample of subjects. Next they use resting state fMRI data to predict these latent factors in a subset of sample from the total sample. This prediction was performed with 10 fold cross validation. This approach has data contamination. In other words, the training data and the testing data have been mixed, because all the data was used to create the latent factors and then it was split into training data and testing data. Data contamination can result in over inflation of prediction performance. To overcome this data contamination, the authors would need to redo this analysis in one of two ways.

- 1) the authors could follow the approach used in Rapuano et al 2020 and calculate the latent factors of executive function in only those individuals without resting state fMRI data. Next the executive function measures in the individuals with resting state fMRI data can then be projected onto these independent latent factors. Finally, connectome based predictive modeling can be used to predict these projections.
- 2) The authors could follow the approach used in Barron et al 2021. First, the subjects with resting state fMRI data would be split into training data and testing data with 10 fold cross validation. Next with in only the training data, the latent factors of executive function can be calculated. Then (as in the previous approach), the executive function measures for the testing data can be projected onto these latent factors with these projections being used for prediction.
- 3) Alternatively, the authors could drop the connectome based prediction analyses altogether and perform more standard explanatory analyses in the whole sample and correct for multiple comparisons using the network base statistic (NBS) or false discovery rate (FDR)

The initial random splitting of the data into the ten folds should be done several times (maybe 100 to 1000 times). When performing a single random split into ten folds, the authors could have simply been lucky (or unlucky) in finding a split the gives highly significant (or un significant) results. Running multiple iterations will allow the authors to understand the variance of predictions.

The DICE overlap analyses is a bit hard to interpret. For example are these overlaps significant? As there is no parametric conversion between DICE and p values, the authors will likely need to use permutation testing, where they randomly the same number of edges (as the "real" results) to networks and compare the overlap between networks. The parametric version of this analysis involves using the Hypergeometric cumulative distribution function(see <https://www.mathworks.com/help/stats/hygecdf.html> for a mathematical description). Other approaches to assess model similarity would be to compute correlations between the regional and brain network contributions to the prediction. The authors can also see if the models cross predict. In other words, does a model trained on one factor of executive function predict a different executive function. Barron et al goes over these approaches to compare different CPM models of latent factors of memory.

The following statement is simply not true “This strategy typically relies on structural MRI, or resting state fMRI because they are less time consuming than task fMRI and thus allow for a larger sample size and leave time for more out-of-scanner behavioral tasks.” Many studies use task fMRI in large samples to do prediction of cognitive traits. See for example Jiang et al for a recent papers that use task fMRI with connectome base predictive model (the same approach used here) in samples with ~1000 subjects.

Rapuano KM, Rosenberg MD, Maza MT, Dennis NJ, Dorji M, Greene AS, Horien C, Scheinost D, Todd Constable R, Casey BJ. Behavioral and brain signatures of substance use vulnerability in childhood. *Dev Cogn Neurosci*. 2020 Dec;46:100878. doi: 10.1016/j.dcn.2020.100878. Epub 2020 Nov 3. Erratum in: *Dev Cogn Neurosci*. 2020 Dec 8;:100891. PMID: 33181393; PMCID: PMC7662869.

Barron DS, Gao S, Dadashkarimi J, Greene AS, Spann MN, Noble S, Lake EMR, Krystal JH, Constable RT, Scheinost D. Transdiagnostic, Connectome-Based Prediction of Memory Constructs Across Psychiatric Disorders. *Cereb Cortex*. 2021 Mar 31;31(5):2523-2533. doi: 10.1093/cercor/bhaa371. PMID: 33345271; PMCID: PMC8023861.

Jiang R, Zuo N, Ford JM, Qi S, Zhi D, Zhuo C, Xu Y, Fu Z, Bustillo J, Turner JA, Calhoun VD, Sui J. Task-induced brain connectivity promotes the detection of individual differences in brain-behavior relationships. *Neuroimage*. 2020 Feb 15;207:116370. doi: 10.1016/j.neuroimage.2019.116370. Epub 2019 Nov 18. PMID: 31751666; PMCID: PMC7345498.

REVIEWER COMMENTS

Reviewer #1 (Remarks to the Author):

I want to reiterate that this paper includes a lot of really interesting analyses and adds to the field. Additionally, the current version is an improvement on the previous submission. A number of the edits that the authors made significantly improved the manuscript.

There are still areas for improvement. The main remaining issues surround overstatement of some conclusions (which can be reframed and/or softened), precision of language for clarity, and further motivation for certain analytic choices.

We again appreciate this reviewer's overall positive evaluation of our revision and his/her valuable suggestions to further improve the manuscript.

1) Some of the conclusions drawn by authors still seem a bit overstated. For example in the section discussing the genetic correlations it is stated, "These results demonstrated that the common EF and shifting-specific components showed dissociable patterns at the genetic level." Yes, their correlation between the common EF and shifting-spec scores was close to zero (as expected - like they point out) however, only one correlation out of all of the others that they ran was significant after correction for multiple testing. While the correlation coefficients were larger for a number of the other pairs, given the lack of significant findings (likely due to imprecision around the estimates) make it hard to draw the conclusion that the lack of correlation between cEF and Shift-spec is notable.

Thanks for this insightful comment. In this study, we found that the genetic correlation between the common EF and shifting-specific components was close to zero ($r = -.06$), much lower than other component pairs, such as the common and updating-specific pair ($r = .61$), and the common and inhibiting-specific component pair ($r = .70$). It is worth noting this genetic dissociation could not be simply attributed to the low heritability of the shifting-specific component, as the updating-specific and inhibiting-specific components also showed low and non-significant heritability. Furthermore, our brain expression-based gene-sets enrichment analysis showed double dissociation for the genetic basis of the common EF and shifting-specific components.

We also agree with this reviewer that the heritability of shifting-specific, updating-specific and inhibition-specific components were not statistically significant, likely due to the limited sample size. This might limit our ability to draw firm conclusions. As a result, we have rephrased this sentence (lines 506~507), which reads:

"These results suggested genetic dissociations between the common EF and shifting-specific components."

Additionally, the conclusion that the cEF+shi-spe model was the best fitting model overall is too strongly stated. It was not the best fitting behavioral model. There was some evidence for it being a compelling neural model, which then they carried forward as the only model for some of their subsequent analyses. For any of those analyses, it is misleading to say it was the best fitting model, because they did not compare it to others. Additionally, when they did compare it to other models, it may have performed better for technical reasons that have been brought up in prior comments. The authors state that the models including a separate component for updated are "not fully compatible with all the cognitive, neural, and genetic data." However it was consistent with the cognitive data and was rarely tested in the genetic data. For example the evidence for genetic influence on shifting was as weak, if not weaker than updating in the gcta heritability analyses. The only other time they look at updating in relation to any of their genetic analyses is with the genetic correlations, and I discuss that previously.

If the authors were to reframe this section to focus on the fact that there were converging lines of evidence that suggest that the cEF and shifting-specific components may be neurally and genetically distinct, that would be well supported by these sets of analyses. However the focus on "best model" seems somewhat distracting from the most interesting parts of this paper.

We thank this reviewer for these comments. As we have stated in the first section of the discussion, we found several lines of evidence that suggest that the “C+S” model showed an overall robust fit to the behavioral, neural and genetic data. We think this convergent evidence is important when examining the ontology of complex cognitive traits.

In our revised manuscript, we reanalyzed the data following the suggestions of Reviewer #3 and directly compared the five candidate models for our neural and genetic analyses. The neural data suggest that only the components in the “C+S” model could be successfully predicted by the RSFC and were associated with distinctive neural substrates. At the genetic level, we found that the components in the “C+S” model showed the lowest genetic correlates, and further enrichment analysis confirmed the double dissociations in the enrichment pattern between the two components, i.e., the MFG-related genes showed significant enrichment for the common EF component, but not for the shifting-specific component. By contrast, the SCG-related genes were significantly enriched for the shifting-specific component but not for the common EF component.

Nevertheless, we completely understand the concerns raised by this reviewer and Reviewer #2. Following their suggestions, we have rephased our discussion to emphasize the convergent evidence from behavioral, neural and genetic data showing the dissociations of common and shifting-specific components, which support the unity and divergent model of EF. The new text reads:

“The convergent evidence for the “Common +Shifting-specific” bifactor model

We obtained convergent evidence from the behavioral, neural, and genetic data to support the “C+S” bifactor model and its “unity and diversity” structure of EF.”

2) Precision in language - at times when the authors are comparing the different models, it's hard to tell if they're referring to updating or shifting "components" from the correlated model or if they really mean updating-specific and shifting-specific cognitive processing from the bifactor models. Therefore, it would be useful to consistently refer to the updating- and shifting-specific abilities as such, and then continue to refer to the updating, inhibiting, or shifting abilities as they currently do.

Thanks for this suggestion. In this revision, we have consistently used the updating-, inhibiting-, and shifting-specific factor to refer to the components in the bifactor models, and the updating, inhibiting, or shifting factor to refer to the components in the correlated model.

Phenotypic Factor Model results:

Best fitting model: Common EF + Updating-spec + Shifting-spec

Neural Data:

Best fitting model: Common EF + Shifting-specific

Neurosynth Section:

The authors clarify that their neural data is based-on individual differences and the neurosynth maps are based on group averages. However, it is not directly addressed that areas important for individual differences in performance do not necessarily map onto the regions that are used in general (or on average) to complete a task. While the overlapping regions show some sort of convergence that those areas are important for those abilities (and possibly adds validity to the CPM methods), it would be useful to point out that.

Thanks for this suggestion. We have emphasized this point in the Results section (lines 360–364), which reads:

“Because CPM and Neurosynth meta-analysis capture different types of neural correlates of EF (i.e., correlates based on individual differences and those based on group-level analysis, respectively), the conjunction analysis provided a more robust and convergent examination of brain regions important for EF and its components.”

Node Enrichment Section:

It is unclear to me why this was done instead of conducting a more typical resting-state network analysis. It could be because this has already been done in other samples, however it would be useful to address what this tells us above and beyond those results.

In the literature, a common approach in resting-state network analysis is to correlate node activity (such as reho, alff or SD) and/or connectivity with behavioral performance.

Although this approach is simple and straightforward, it could lead to unreliable and inflated results, especially when there are so many variables (e.g., edges). In contrast, the CPM used a multivariate cross-validation approach to examine the relation between connectivity strength and cognition, which could overcome the above limitations. As noted by the third reviewer, this approach is increasingly used in the literature (Barron et al., 2021; Jiang et al., 2020; Rapuano et al., 2020). We have added some rationales on the CPM methods and discussed our results in relation to previous findings (lines 675~691), which reads:

“A few previous studies have used resting-state data and calculated simple correlations between resting-state data and behavioral performance to examine the neural correlates of common and specific aspects of EFs. For example, Reineberg et al.(2015) extracted common, updating-specific, and shifting-specific components from three tasks (n=91), and found that the common EF was associated with the connectivity between the frontal pole and the attentional network, and that between the cerebellum and the right frontoparietal network. In contrast, the shifting-specific ability was associated with the connectivity between the angular gyrus and the ventral attention network. In a follow-up study using a large sample (n = 250) and six EF tasks, they found that the shifting-specific component was correlated with the connectivity between the ventral attention network, particularly the cingulo-opercular subsystem, and the default mode network(Reineberg, Gustavson, Benca, Banich, & Friedman, 2018). Using a larger sample and a cross-validation approach, we found that the contributing edges linked to the MFG for the common component were mainly located in the FPN-SAN network, whereas the contributing edges linked to the paracingulate gyrus for the shifting-specific component were mainly located in the CON-Subcor network. These results together suggest that the fronto-parietal network may play a key role in common EF, whereas the cingulo-opercular subsystem may support shifting-specific EF.”

AHBA section:

*Did you test other thresholds other than $\pm 50\text{kb}$? $\pm 35\text{kb}$ has recently been frequently cited in the literature (Wu, Y., Zheng, Z., Visscher, P. M. & Yang, J. Quantifying the mapping precision of genome-wide association studies using whole-genome sequencing data. *Genome Biol* 18, (2017))*

Thanks for this suggestion. For the enrichment analysis, we have done an additional analysis using the $\pm 35\text{kb}$ threshold, and found that the MFG-related genes showed significant enrichment for the common EF component ($p_{75\text{percentile}} = 1.80 \times 10^{-2}$, FDR-BH corrected, across 12 tests), but not for the shifting-specific component ($p_{75\text{percentile}} = .08$, FDR-BH corrected, across 12 tests). In contrast, the SCG-related genes were significantly enriched for the shifting-specific component ($p_{75\text{percentile}} = 1.80 \times 10^{-2}$, FDR-BH corrected, across 12 tests), but not for the common EF component ($p_{75\text{percentile}} = .18$, FDR-BH corrected, across 12 tests).

Furthermore, we did not find any significant results for the LOC-related and BG-related genes (for the updating-specific component, see below for our response to the comment from Reviewer #2). These two gene sets were not included in further analyses.

In this revision, we have updated the results based on this new analysis.

In a later section you state that you used: 50kb (or 25kb, 75kb, 100kb), however it seems more pertinent that you use smaller windows rather than larger (e.g., Palmer RHC, Benca-Bachman CE, Huggett SB, et al. Multi-omic and multi-species meta-analyses of nicotine consumption. Transl Psychiatry. 2021;11(1):98. Published 2021 Feb 4. doi:10.1038/s41398-021-01231-y)

Thanks for this suggestion. In addition to the 35kb results reported in the main text, we have also reported the result from ± 25 and ± 50 kb in Table S5.

*The genetic architecture of common and shifting-specific components based on psychiatric or cognitive associated gene sets:
Please motivate the use of this enrichment method (rather than using polygenic scores to predict the cEF and Shifting-specific scores). What does this analysis tell us that a PGS would does not?*

Following this reviewer's suggestion, we also used a standard approach (i.e., clumping and thresholding) implemented in PLINK (Purcell et al., 2007) to calculate polygenic score ($n=1454$), which is defined as the sum of allele counts (genotypes), weighted by estimated effect sizes (i.e., beta coefficients or odds ratio) obtained from GWAS studies (i.e., SCZ, ADHD, IQ, EA, Crohn's disease). This method is widely used in the literature (Dudbridge, 2013; Euesden, Lewis, & O'Reilly, 2015; Wray, Goddard, & Visscher, 2007; Wray et al., 2014).

Two filtering steps have been applied in the clumping and thresholding (C+T) method. First, when there are multiple significant association variants in the same region (e.g., within 250kb distance), the clumping (C) procedure keeps the most significant SNP if these SNPs show linkage disequilibrium (i.e., r -squared of 0.50). Second, thresholding (T) removes variants with a p value larger than a chosen level of significance (i.e., $p > .05$, $p > .01$, $p > .001$). We estimated the partial correlations between polygenic scores and EF component scores, with age, gender, genotype array, site and population stratification (top 10 PCs) as covariates.

We found no significant results at any threshold, except the correlation between intelligence-related genes and the common EF component ($r = .08$, $p = .022$, FDR-BH corrected, across 10 tests, threshold $p=.05$). Because of the non-significant result, we did not include this analysis in this revision.

Reviewer #2 (Remarks to the Author):

This is a revision of a manuscript for which I was a reviewer when it was submitted to a different Nature Journal. Overall, I found the authors very responsive to the prior reviews. However, I do still have some concerns about the primary conclusion and some remaining questions about details. I detail these concerns and questions in the following paragraphs, but the bottom line is that overall, I think this is an impressive study that will be of great interest to the field, and I believe my concerns could be addressed with another revision.

We thank this reviewer for acknowledging our effort in revision and his/her valuable suggestions to further improve the manuscript.

Most importantly, I am glad that the authors have added the discussion of the weak updating-specific results and limitations in the discussion, but I remain uncomfortable with some of the statements in the text. In particular, the authors continue to characterize the C+S model as the “best-fit model” (line 682). I continue to believe it is misleading to characterize this as the “best fit” model for several reasons:

First, it is statistically worse than models with an updating-specific component in terms of chi-square difference tests, and shows large enough change in CFI/RMSEA to warrant the conclusion that it is providing poorer fit than the C+U+S model (which is also supported in the prior literature) or the C + I + S model. For example, studies examining model comparisons in the context of invariance (e.g., <https://doi.org/10.1080/10705510701301834>) have used a criterion of change in CFI > .01 and change in RMSEA > .015.

This is a very good point. We have provided AIC and BIC values in model fitting and the change in CFI/RMSEA during model comparison. Both suggest that the “C+U+S” model has the best fit to the behavioral data. We have acknowledged this fact in the discussion.

Second, some of the comparisons the authors make based on the neural and genetic data are inconsistent. For example, on line 381: “Taken together, the above results suggested that the “C+S” model was the best-fit model based on the neural data, because its shifting component was predicted with greater accuracy than was the shifting component in the “U/I+S” model, and because it had a greater degree of dissociated edges for each component than did the “U/I+S” model or the “I+U+S” model. These results indicate that there are specific neural substrates for the shifting component, but the neural bases of the inhibiting and updating components are likely to be highly overlapping.” I don’t disagree that the neural components for U and I are likely to be overlapping, but this conclusion that the C+S model is best seems unwarranted given that the comparisons the authors are making were not tested for significant differences and ignore some non-differences (the shifting component seems equally well predicted in the C+I+S, C+U+S, and C+S models, so why only focus on comparing the C+S to the correlated factors models?). Moreover, by this logic, the C+U+S or C+I+S models should be preferred because they show better prediction of C than the

C+S model.

As suggested by Reviewer #3, we have done a couple of analyses to test the significance of the neural overlaps. In addition, we have rephrased our conclusion following this reviewer and Reviewer #1's suggestion. That is, instead of emphasizing that the "C+S" is the best model, we conclude that the "C+S" model is supported by convergent behavioral, neural and genetic evidence.

With respect to the genetic data, these genetic analyses are very interesting, but the authors don't really comment until late in the discussion on the low, non-significant heritability of the shifting component across models. As the authors note, it may be just a power issue, given that twin studies have found lower heritability for shifting components, but it does beg the question of how these results are constraining model selection, given that in this sample there is little evidence for unique genetic variance in shifting, at least at the genome-wide level. Of course, it makes sense that the authors don't put too much weight on this point given that this sample size has very low power for these kinds of analyses, yet it does present an inconsistency in terms of how they are using the results to constrain their model selection.

Thanks for this insightful comment. As stated in our response to the first reviewer's comment #1, we found that the genetic correlation between the common EF and shifting-specific components was close to zero ($r = -.06$), much lower than other component-pairs, such as common and updating-specific ($r = .61$), common and inhibiting-specific component ($r = .70$). It is worth noting this genetic dissociation could not be simply attributed to the low heritability of the shifting-specific component, as the updating-specific and inhibiting-specific components also showed low and nonsignificant heritability. Furthermore, our brain expression-based gene-sets enrichment analysis showed double dissociation for the genetic basis of the common EF and shifting-specific components.

We agree with this reviewer that the heritability of shifting-specific, updating-specific and inhibition-specific components was not statistically significant, likely due to the limited sample size. This might limit our ability to draw firm conclusions. As a result, we have softened our conclusion and rephrased our discussion (see below).

*Third, and more generally, I worry about the claim that this limited set of neural data can constrain model selection. Perhaps it can, but my concern is that the authors have only examined resting state data. Although I think it would be a lot to try to add structural measures to the analyses, it would be worth discussing whether other neural measures may have resulted in different conclusions. The authors justify their use of RSFC as "optimal intermediate phenotypes" (line 313), but in prior studies resting state does not always predict outcomes strongly and generally shows lower reliability and heritability compared to structural brain measures, as illustrated by some of the citations I listed in my prior review. In fact, prior studies examining brain predictors of the unity/diversity model specifically (which were cited by the authors here) did not find RSFC measures that predicted the updating-specific component (Reineberg et al., 2015), but *did* find*

**structural* correlates of updating-specific (Smolker et al., 2015; 2018). Thus, it could be that the conclusion that the updating-specific component is only weakly discriminated is specific to resting state data.*

Thanks for this suggestion. Two studies have examined the anatomic basis for the different components in the “C+U+S” model (Smolker, Depue, Reineberg, Orr, & Banich, 2015; Smolker, Friedman, Hewitt, & Banich, 2018). One study found that the updating-specific component was related to the dorsolateral prefrontal gray matter volume (Smolker et al., 2015). However, this study had a small sample size (n =68), and only used one task for each EF component, i.e., category-switch task, anti-saccade task and keep track task. Thus, their results might reflect task-specific processes rather than the latent updating component. A follow-up study (Smolker et al., 2018) used a larger sample (n= 251) and six EF tasks. The anti-saccade, category-switch, and keep track tasks were completed during an fMRI session immediately following the T1 structural scan. The Stroop, letter memory, and number-letter tasks were conducted as part of a larger behavioral battery immediately after the scanning session. They found that better updating-specific ability was associated with greater cortical thickness of a cluster in left cuneus/precuneus, and reduced cortical thickness in regions of right superior frontal gyrus and right middle/superior temporal gyrus, but no significant results for white matter diffusion.

In this revision, we have added more discussion on the anatomical results and call for future studies to examine the neuroanatomic basis of EF components (lines 731~739), which reads:

“It is also notable that previous studies have revealed mixed results for the anatomical basis of the updating-specific component. For example, one study found that the updating-specific component was related to the dorsolateral prefrontal gray matter volume(Smolker et al., 2015). However, that study had a small sample size (n = 61) and used only one task for each EF component, so its results might reflect task-specific processes rather than the latent updating-specific component. A follow-up study(Smolker et al., 2018) with a larger sample size (n = 251) and six EF tasks found that better updating-specific ability was associated with greater cortical thickness of a cluster in the left cuneus/precuneus, and reduced cortical thickness in the right superior frontal gyrus and right middle/superior temporal gyrus.”

Some of the discussion is selective about citing supporting evidence for dissociating shifting, ignoring evidence that might speak to the validity of the updating-specific component. For example, Line 748: “Fourth, a previous study found that the self-restraint ability was significantly associated with the common EF ability at age 17, but not with the updating-specific ability⁴¹, suggesting that the updating-specific might not make unique contributions to the self-restraint beyond the common component.” I don’t follow why the authors are focusing on this one study here. Friedman and Miyake (2017) discussed how different outcomes show different profiles of relationships with the EF components, such that updating-specific is generally not related to psychopathology

or self-restraint, but is related to intelligence, whereas psychopathology and self-restraint is most closely related to the common component, with only small negative relationships to the shifting-specific component for some behaviors. So by this logic, the validity of the updating-specific component is strong given this prior work suggesting discriminant validity.

We have revised the discussion in this section (lines 717~724), which reads:

“Third, the cognitive processes tapped by the updating-specific component are not yet clear. Both our study and the previous studies(Friedman, Miyake, Robinson, & Hewitt, 2011; Friedman et al., 2008) found that the updating-specific component was moderately correlated with intelligence beyond the common component, but it did not predict self-restraint ability(Friedman et al., 2011), or procrastination(Gustavson, Miyake, Hewitt, & Friedman, 2015), beyond the common EF. It has been proposed that the updating-specific component might be involved in effective gating of information and controlled retrieval from long-term memory(Miyake & Friedman, 2012). More studies are required to examine the cognitive processes of the updating-specific component.”

Moreover, many of the follow-up models (e.g., using neurosynth, gene sets, etc.) focus on the C and S components in the C+S model, but some of the same analyses could be extended to include the U component in the bifactor models. I would be interested to see what neural and genetic areas may be suggested by the neurosynth maps and gene pathways, if any.

This is a very good point. In response to this reviewer’s comment, we have obtained the updating-specific meta-analytic map (using neurosynth data) by subtracting the common activation regions of the three meta-analytic maps from the updating meta-analytic map, which revealed a distributed neural network, including the MFG, frontal pole, occipital fusiform cortex, planum temporale, posterior cingulate gyrus, precuneus cortex, right thalamus and basal ganglia. It should be emphasized that not all these regions revealed by neurosynth meta-analysis are critically involved in the updating-specific component. Since the CPM analysis did not reveal significant prediction of the updating-specific component, it is thus hard to use conjunction analysis to examine the nodes for the updating-specific component.

In addition to this data-driven approach, we also tried the hypothesis-driven approach. In particular, previous studies have implicated the basal ganglia (BG) for the updating-specific component brain region (Frank MJ, Loughry B, & RC., 2001; Friedman & Miyake, 2017). Consistently, the basal ganglia is one of the top clusters of meta-analytic results of the updating-specific component ($MNI_{(COG)} = 13, -3, 4$). In the first analysis, we examined whether the edges connecting the basal ganglia could predict the updating-specific component, which revealed non-significant result ($r = -.015$, $p_{permutation}=.60$, uncorrected).

In addition, we extracted genes with enhanced expression in the basal ganglia region using AHBA. Our results indicated that the basal ganglia genes did not show any enrichment for the updating-specific component ($p_{75\text{percentile}} = .53$, FDR-BH corrected, across 12 tests). We also estimated the enrichment patterns of other three candidate gene sets for the updating-specific component, which also failed to reveal any significant results (MFG-related genes: $p_{75\text{percentile}} = .32$, FDR-BH corrected, across 12 tests; SCG-related genes $p_{75\text{percentile}} = .27$, FDR-BH corrected, across 12 tests; LOC-related genes $p_{75\text{percentile}} = .27$, FDR-BH corrected, across 12 tests). Similar results were found with other genetic boundaries (25kb, 50kb).

Together, the additional analysis still failed to reveal significant neural and genetic correlations for the updating-specific component. We have briefly mentioned the null result in this revision.

All this said, I do still think this is an impressive study and set of analyses, and I am glad to see how consistent the results are in this sample vs. others at a general level. I am not necessarily suggesting that the authors add substantial analyses (such as adding structural data), although examining some of the neurosynth and genetic results for the updating specific factor would be appreciated. I believe that the authors could address my primary concerns by removing remaining references to “best fit” models. Perhaps rather than framing the goal as selecting the best EF model, the authors could frame this more as a characterization of the unity and diversity components in terms of RSFC and genomic variance, with the conclusion about the updating-specific component being that it is relatively weaker, at least in terms of the sample and metrics examined here.

Thanks for this very encouraging and constructive suggestion. Considering both this reviewer and Reviewer #1's comments, we have now added more analyses to examine the neural and genetic bases of the updating-specific component (see above). In addition, we have rephrased the first section of the discussion to emphasize the unity and diversity structure of EF, as supported by the behavioral, neural and genetic dissociation. In particular, the subtitle of this section is now changed to **“The convergent evidence for the “Common +Shifting-specific” bifactor model”**

And the first sentence of this section is changed to “We obtained convergent evidence from the behavioral, neural, and genetic data to support the “C+S” bifactor model and its unity and divergent structure of EF.”

Furthermore, we have added more discussion on the updating-specific component by calling for more studies to examine the cognitive, neural and genetic bases of updating-specific components (see our response above).

Specific comments:

Results:

Given that both nested and non-nested models are compared, it would be useful to include information criteria (AIC and BIC) in addition to the CFI, RMSEA, and SRMR. In addition, nested models can be compared with chi-square difference tests and change

in CFI/RMSEA.

We have included the information in this revision.

I was also quite surprised at the high h^2_{SNP} , as I believe those are much stronger than typical reports for cognitive measures. Have the authors calculated heritability of their IQ measure to see if it is also higher than prior reports?

We did the GCTA heritability analysis on IQ, which revealed a heritability of $h^2_{SNP} = .50$ (SE = .38, $p = .10$), in line with a previous meta-analysis based on 50 years of twin studies ($h^2 = .54$) (Polderman et al., 2015). This value is higher than that found in a prior consortium study based on unrelated individuals (Savage et al., 2018) i.e., $h^2_{SNP} = .19\sim.22$ across age groups; $h^2_{SNP} = .22$, SE=.10 for young adults (n=6033). We have reported this result and provided a plausible explanation of our higher heritability (lines 483~491). Which reads:

“We also estimated the heritability of IQ (Raven’s Progressive Matrices Test), which revealed a heritability of $h^2_{SNP} = .50$ (SE = .38, $p = .10$), in line with a previous meta-analysis based on 50 years of twin studies ($h^2 = .54$)(Polderman et al., 2015). This value is higher than that found in a prior consortium study based on unrelated individuals(Savage et al., 2018) i.e., $h^2_{SNP} = .19\sim.22$ across age groups; $h^2_{SNP} = .22$, SE=.10 for young adults (n=6033). The higher heritability estimates may be due to the homogeneity of our sample of healthy young college students of Han ethnicity. Using the same sample, we also found higher heritability estimates of functional connectivity edges in a previous study (Feng et al., 2020) as compared to those from the Biobank data(Elliott et al., 2018).”

Discussion:

Line 723: “It is worth noting this genetic dissociation should not be purely contributed to low heritability for the shift-specific component, as the updating-specific and shift-specific components also showed low and insignificant heritability.” Perhaps the authors meant to say “the updating -specific component also showed low and insignificant heritability”?

Thanks for noting the error. We have made the correction (as well as some grammatical errors).

Method:

Thank you for adding more information about the methods, but I did still have some questions:

Please clarify the total number of trials for the Stroop task, the switch tasks, and the keep track task.

We have added more details in this revision. In particular, the numbers of trials for these tasks were provided below:

Color-Word Stroop: 96 trials
Keep track: 180 trials
Number-letter switching task: 160 trials
Color-shape switching task: 160 trials
Category switching task: 96 trials

What was the response to cue interval for the switch tasks?

The cue was presented for 150ms before the target for the “color-shape” and category switching tasks. No cue was presented for the number-letter switching task. The participants did not need to respond to cue.

The “letter memory” task is really a 3-back task; it should not be called letter memory to avoid confusion with prior studies.

It has been changed to letter 3-back task.

Line 985: “The stimulus disappeared until participants made a response.” I think the authors mean that the stimulus disappeared after the participants made a response?

This has been corrected.

Please clarify how internal consistency was computed (e.g., one could use odd/even trials, correlations between blocks, etc., and were correlations corrected with the Spearman-Brown prophecy formula)? It’s also not clear why consistency could not be computed for stop-signal and keep track; for stop-signal the SSRT can be calculated for each block and a Cronbach’s alpha estimated; for keep track prior studies have used a Cronbach’s alpha computed on subsets of trials (e.g., the average of the first trials at each difficulty level, the average of the second trials at each difficulty level, and so on).

We used odd/even trials and the correlations were corrected with the Spearman-Brown prophecy formula.

For the stop-signal task, we used four stair-cases with different starting values, which were randomly mixed together. Thus, it is hard for us to separate them into odd and even trials. We therefore decided to calculate the SSRT for the last two blocks. 391 subjects had at least one inflection point for one of the four staircases in both blocks, and the correlation was .55 (corrected with Spearman-Brown prophecy formula).

For keep track task, we calculated the internal consistency across 4 sets of trials at each difficult level following this reviewer’s suggestion, which was .67.

These new results have been added in this revision (Table S1).

Line 1109: *“To build the predictive model, we summarized, for each participant in the training set, the selected edges’ strength values to a single connectivity value and used it to predict the EF factor score with a linear regression.*

” Please clarify what “summarized” means. One could add the strength values with unit weighting, do a weighted sum; moreover, how were positive vs. negative predictions treated?

We used unit weighting for both positive and negative edges. We have modified this statement (lines 1069~1074), which reads:

“We calculated four unit-weighted summary scores of the connectivity strength values: all selected positive edges and all selected negative edges for the training and testing sets separately. To build the predictive model, we used the summary scores of the positive and negative edges from the training set to predict the EF factor score in a linear regression. This predictive model was then used to predict the EF factor score in the testing set.”

Reviewer #3 (Remarks to the Author):

The paper by Feng and co authors, titled “A cognitive neurogenetic approach to uncovering the structure of 2 executive functions”, and submitted to the journal Nature Communications, presents a three fold study of behavior, neuroimaging, and genetics to understand the neurogenetics underlying of executive function. Giving the wide area of study and the previous reviews/reviewers expertise, I focused my review on the neuroimaging aspect of this work. Overall, while there is an impressive amount of effort that went into this study, I find the neuroimaging aspect to have a few methodological flaws, which limit my enthusiasm for the study.

There is a data contamination problem with the cross validation strategy used. The authors first construct latent factors of executive function based on the total sample of subjects. Next they use resting state fMRI data to predict these latent factors in a subset of sample from the total sample. This prediction was performed with 10 fold cross validation. This approach has data contamination. In other words, the training data and the testing data have been mixed, because all the data was used to create the latent factors and then it was split into training data and testing data. Data contamination can result in over inflation of prediction performance. To overcome this data contamination, the authors would need to redo this analysis in one of two ways.

1) the authors could follow the approach used in Rapuano et al 2020 and calculate the latent factors of executive function in only those individuals without resting state fMRI data. Next the executive function measures in the individuals with resting state fMRI data can then be projected onto these independent latent factors. Finally, connectome based predictive modeling can be used to predict these projections.

2) The authors could follow the approach used in Barron et al 2021. First, the subjects with resting state fMRI data would be split into training data and testing data with 10 fold cross validation. Next with in only the training data, the latent factors of executive function can be calculated. Then (as in the previous approach), the executive function

measures for the testing data can be projected onto these latent factors with these projections being used for prediction.

3) Alternatively, the authors could drop the connectome based prediction analyses altogether and perform more standard explanatory analyses in the whole sample and correct for multiple comparisons using the network base statistic (NBS) or false discovery rate (FDR)

Thanks for this very helpful comment. We have redone the CPM following the Rapuano et al (2020) approach. The results were largely unchanged, except that the shifting component in the two correlated models (“U/I+S”, “U+I+S”) was not successfully predicted by the RSFC patterns.

We have included the new results in this revision and added some methodological details (lines 1053~1060), which reads:

“Following Rapuano et al. (2020), to avoid data contamination between the behavioral models and the neuroimaging analysis, component scores were recomputed using the behavioral subset of participants (n = 1240). The resulting component scores were consistent with those obtained when the whole sample was used ($r > 0.9$). The loadings from the behavioral subset of participants were subsequently used to transform EF performance data in the CPM analysis (n = 870). Furthermore, to avoid biasing the test set, edge strengths were standardized (z-scores) across subjects within each fold of the training set and the test set, separately.”

The initial random splitting of the data into the ten folds should be done several times (maybe 100 to 1000 times). When performing a single random split into ten folds, the authors could have simply been lucky (or unlucky) in finding a split that gives highly significant (or un significant) results. Running multiple iterations will allow the authors to understand the variance of predictions.

We have done 100 random splitting of the data, and the results were averaged. This information has been added in the Method section.

The DICE overlap analyses is a bit hard to interpret. For example are these overlaps significant? As there is no parametric conversion between DICE and p values, the authors will likely need to use permutation testing, where they randomly sample the same number of edges (as the “real” results) to networks and compare the overlap between networks. The parametric version of this analysis involves using the Hypergeometric cumulative distribution function (see <https://www.mathworks.com/help/stats/hygecdf.html> for a mathematical description). Other approaches to assess model similarity would be to compute correlations between the regional and brain network contributions to the prediction. The authors can also see if the models cross predict. In other words, does a model trained on one factor of executive function predict a different executive function. Barron et al goes over these approaches to compare different CPM models of latent factors of memory.

We thank the reviewer for the suggestion of performing significance tests. We have used permutation testing to examine the statistical significance and added the new results in this revision (Figure 4). In particular, we randomly selected the same number of contributing edges for each component 10000 times and calculated the Dice similarities of EF components within the same models. We found a significant overlap of the contributing edges between the inhibiting and updating components in the “I+U+S” model (Dice coefficient = .58, $p_{\text{permutation}} < 1 \times 10^{-4}$, uncorrected) (Figure 4A). In contrast, the overlap of the contributing edges between the common and shifting-specific components in the “C+I+S” (Figure 4B), “C+U+S” (Figure 4C) and the “C+S” bifactor model (Figure 4D) (Dice coefficients range from .007 to .014, $p_{\text{permutation}} > .52$, uncorrected) were non-significant, suggesting dissociated neural substrates for the common and shifting-specific components.

The following description of the permutation procedure has been added to the revised manuscript (lines 1095~1102). Which reads:

“We examined the statistical significance of a Dice coefficient using 10000 resampled data sets, in each of which we randomly selected |X| and |Y| edges for the first and second components, respectively, and used them to calculate the Dice similarity coefficient. Note that in each resampled data set, each edge was either selected or unselected for each component. Thus, this procedure is equivalent to permutating the edge selection status 10000 times for each component. The resulting 10000 Dice coefficients were used to construct the null distribution. Multiple comparisons correction was performed using FDR(Benjamini & Hochberg, 1995).”

The following statement is simply not true “This strategy typically relies on structural MRI, or resting state fMRI because they are less time consuming than task fMRI and thus allow for a larger sample size and leave time for more out-of-scanner behavioral tasks.” Many studies use task fMRI in large samples to do prediction of cognitive traits. See for example Jiang et al for a recent papers that use task fMRI with connectome base predictive model (the same approach used here) in samples with ~1000 subjects.

Thanks for raising this issue. We have modified this statement (lines103~109), which reads:

“This strategy typically relies on structural MRI (Smolker et al., 2015; Tamnes et al., 2010) or resting-state fMRI (Reineberg et al., 2015; Reineberg et al., 2018), and sometimes task fMRI with a few tasks (Greene, Gao, Scheinost, & Constable, 2018; Jiang et al., 2020), because they are less time consuming and more cost efficient than scanning all tasks and thus allow for a larger sample size and leave time for more out-of-scanner behavioral tasks. Nevertheless, most studies usually used a small number of behavioral tasks and/or a small sample. More critically, they used neural data only to verify their pre-defined model of mental structure rather than to assess several candidate models.”

Rapuano KM, Rosenberg MD, Maza MT, Dennis NJ, Dorji M, Greene AS, Horien C, Scheinost D, Todd Constable R, Casey BJ. Behavioral and brain signatures of substance use vulnerability in childhood. *Dev Cogn Neurosci*. 2020 Dec;46:100878. doi: 10.1016/j.dcn.2020.100878. Epub 2020 Nov 3. Erratum in: *Dev Cogn Neurosci*. 2020 Dec 8;:100891. PMID: 33181393; PMCID: PMC7662869.

Barron DS, Gao S, Dadashkarimi J, Greene AS, Spann MN, Noble S, Lake EMR, Krystal JH, Constable RT, Scheinost D. Transdiagnostic, Connectome-Based Prediction of Memory Constructs Across Psychiatric Disorders. *Cereb Cortex*. 2021 Mar 31;31(5):2523-2533. doi: 10.1093/cercor/bhaa371. PMID: 33345271; PMCID: PMC8023861.

Jiang R, Zuo N, Ford JM, Qi S, Zhi D, Zhuo C, Xu Y, Fu Z, Bustillo J, Turner JA, Calhoun VD, Sui J. Task-induced brain connectivity promotes the detection of individual differences in brain-behavior relationships. *Neuroimage*. 2020 Feb 15;207:116370. doi: 10.1016/j.neuroimage.2019.116370. Epub 2019 Nov 18. PMID: 31751666; PMCID: PMC7345498.

References

- Barron, D. S., Gao, S., Dadashkarimi, J., Greene, A. S., Spann, M. N., Noble, S., . . . Scheinost, D. (2021). Transdiagnostic, Connectome-Based Prediction of Memory Constructs Across Psychiatric Disorders. *Cereb Cortex*, 31(5), 2523-2533. doi:10.1093/cercor/bhaa371
- Benjamini, Y., & Hochberg, Y. (1995). controlling the false discovery rate a practical and powerful approach to multiple testing. *JOURNAL OF THE ROYAL STATISTICAL SOCIETY.Series B(Methodological)*, 57(1), 289-300.
- Dudbridge, F. (2013). Power and predictive accuracy of polygenic risk scores. *PLoS Genet*, 9(3), e1003348. doi:10.1371/journal.pgen.1003348
- Elliott, L. T., Sharp, K., Alfaro-Almagro, F., Shi, S., Miller, K. L., Douaud, G., . . . Smith, S. M. (2018). Genome-wide association studies of brain imaging phenotypes in UK Biobank. *Nature*, 562(7726), 210-216. doi:10.1038/s41586-018-0571-7
- Euesden, J., Lewis, C. M., & O'Reilly, P. F. (2015). PRSice: Polygenic Risk Score software. *Bioinformatics*, 31(9), 1466-1468. doi:10.1093/bioinformatics/btu848
- Feng, J., Chen, C., Cai, Y., Ye, Z., Feng, K., Liu, J., . . . Xue, G. (2020). Partitioning heritability analyses unveil the genetic architecture of human brain multidimensional functional connectivity patterns. *Hum Brain Mapp*, 41(12), 3305-3317. doi:10.1002/hbm.25018
- Frank MJ, Loughry B, & RC., O. R. (2001). Interactions between frontal cortex and basal ganglia in working memory: a computational model. . *Cogn Affect Behav Neurosci*, 1, 137-160. doi:10.3758/cabn.1.2.137.
- Friedman, N. P., & Miyake, A. (2017). Unity and diversity of executive functions: Individual differences as a window on cognitive structure. *Cortex*, 86, 186-204. doi:10.1016/j.cortex.2016.04.023
- Friedman, N. P., Miyake, A., Robinson, J. L., & Hewitt, J. K. (2011). Developmental trajectories in toddlers' self-restraint predict individual differences in executive functions 14 years

- later: a behavioral genetic analysis. *Dev Psychol*, 47(5), 1410-1430.
doi:10.1037/a0023750
- Friedman, N. P., Miyake, A., Young, S. E., DeFries, J. C., Corley, R. P., & Hewitt, J. K. (2008). Individual differences in executive functions are almost entirely genetic in origin. *J Exp Psychol Gen*, 137(2), 201-225. doi:10.1037/0096-3445.137.2.201
- Greene, A. S., Gao, S., Scheinost, D., & Constable, R. T. (2018). Task-induced brain state manipulation improves prediction of individual traits. *Nat Commun*, 9(1), 2807. doi:10.1038/s41467-018-04920-3
- Gustavson, D. E., Miyake, A., Hewitt, J. K., & Friedman, N. P. (2015). Understanding the cognitive and genetic underpinnings of procrastination: Evidence for shared genetic influences with goal management and executive function abilities. *J Exp Psychol Gen*, 144(6), 1063-1079. doi:10.1037/xge0000110
- Jiang, R., Zuo, N., Ford, J. M., Qi, S., Zhi, D., Zhuo, C., . . . Sui, J. (2020). Task-induced brain connectivity promotes the detection of individual differences in brain-behavior relationships. *Neuroimage*, 207, 116370. doi:10.1016/j.neuroimage.2019.116370
- Miyake, A., & Friedman, N. P. (2012). The Nature and Organization of Individual Differences in Executive Functions: Four General Conclusions. *Curr Dir Psychol Sci*, 21(1), 8-14. doi:10.1177/09637214111429458
- Polderman, T. J. C., Benyamin, B., de Leeuw, C. A., Sullivan, P. F., van Bochoven, A., Visscher, P. M., & Posthuma, D. (2015). Meta-analysis of the heritability of human traits based on fifty years of twin studies. *Nat Genet*, 47(7), 702-+. doi:10.1038/ng.3285
- Purcell, S., Neale, B., Todd-Brown, K., Thomas, L., Ferreira, M. A., Bender, D., . . . Sham, P. C. (2007). PLINK: a tool set for whole-genome association and population-based linkage analyses. *Am J Hum Genet*, 81(3), 559-575. doi:10.1086/519795
- Rapuano, K. M., Rosenberg, M. D., Maza, M. T., Dennis, N. J., Dorji, M., Greene, A. S., . . . Casey, B. J. (2020). Behavioral and brain signatures of substance use vulnerability in childhood. *Dev Cogn Neurosci*, 46, 100878. doi:10.1016/j.dcn.2020.100878
- Reineberg, A. E., Andrews-Hanna, J. R., Depue, B. E., Friedman, N. P., & Banich, M. T. (2015). Resting-state networks predict individual differences in common and specific aspects of executive function. *Neuroimage*, 104, 69-78. doi:10.1016/j.neuroimage.2014.09.045
- Reineberg, A. E., Gustavson, D. E., Benca, C., Banich, M. T., & Friedman, N. P. (2018). The Relationship Between Resting State Network Connectivity and Individual Differences in Executive Functions. *Front Psychol*, 9, 1600. doi:10.3389/fpsyg.2018.01600
- Savage, J. E., Jansen, P. R., Stringer, S., Watanabe, K., Bryois, J., de Leeuw, C. A., . . . Posthuma, D. (2018). Genome-wide association meta-analysis in 269,867 individuals identifies new genetic and functional links to intelligence. *Nat Genet*, 50(7), 912-919. doi:10.1038/s41588-018-0152-6
- Smolker, H. R., Depue, B. E., Reineberg, A. E., Orr, J. M., & Banich, M. T. (2015). Individual differences in regional prefrontal gray matter morphometry and fractional anisotropy are associated with different constructs of executive function. *Brain Struct Funct*, 220(3), 1291-1306. doi:10.1007/s00429-014-0723-y
- Smolker, H. R., Friedman, N. P., Hewitt, J. K., & Banich, M. T. (2018). Neuroanatomical Correlates of the Unity and Diversity Model of Executive Function in Young Adults. *Front Hum Neurosci*, 12, 283. doi:10.3389/fnhum.2018.00283
- Tamnes, C. K., Ostby, Y., Walhovd, K. B., Westlye, L. T., Due-Tønnessen, P., & Fjell, A. M. (2010). Neuroanatomical correlates of executive functions in children and adolescents: a

- magnetic resonance imaging (MRI) study of cortical thickness. *Neuropsychologia*, 48(9), 2496-2508. doi:10.1016/j.neuropsychologia.2010.04.024
- Wray, N. R., Goddard, M. E., & Visscher, P. M. (2007). Prediction of individual genetic risk to disease from genome-wide association studies. *Genome Res*, 17(10), 1520-1528. doi:10.1101/gr.6665407
- Wray, N. R., Lee, S. H., Mehta, D., Vinkhuyzen, A. A., Dudbridge, F., & Middeldorp, C. M. (2014). Research review: Polygenic methods and their application to psychiatric traits. *J Child Psychol Psychiatry*, 55(10), 1068-1087. doi:10.1111/jcpp.12295

REVIEWER COMMENTS

Reviewer #1 (Remarks to the Author):

Overall the authors have done a nice job of addressing the majority of my concerns.

There are a few relatively minor revisions that should be made before publication.

1. Stats should be $p < 1 \times 10^{-3}$ rather than just 10^{-3} as in “(r = .27, $p < 10^{-3}$, uncorrected)”
2. The information presented in lines 366-381 (associated brain regions and their MNI coordinates) would be better presented in table format if feasible.
3. Please specify which GCTA tool was used to calculate the genetic correlations.
4. Related, the standard errors should be reported along with the genetic correlations and p values.
5. Please specify sources for the genes related to the analyses for schizophrenia, adhd, educational attainment, etc.

Reviewer #2 (Remarks to the Author):

Overall, the authors have been very responsive to reviewer comments, and I believe they have addressed my comments in this revision. I have only a few minor comments/suggestions that came up in response to some of the additions:

First, the result the SNP-h² for IQ is comparable to that for twin studies is striking, as typically the SNP-h² is ~1/2 the twin h² (doi: 10.1038/nrg.2017.104). Although the authors state that “Using the same sample, we also found higher heritability estimates of functional connectivity edges in a previous study⁴⁹ as compared to those from the Biobank data⁵⁹” (p. 22), the discrepancies there are much less pronounced, as ref 49 reports SNP-h² < 10%. (Also, I think the authors mean “the UK Biobank data”?) In contrast, the SNP-h² for EFs and IQ are in the 50-60% range here, and I’m not sure that could be explained by sample homogeneity alone. That the SNP-h² are so high raises some concerns that something may be different in this analysis, making it less comparable to the reports in the existing literature. However, I had difficulty understanding what that might be when I re-read the methods with this in mind. The methods specify that “To make sure our participants were unrelated and had a similar genetic background (e.g., Han Chinese), we estimated the genetic relationship for each pair of participants using the principal component analysis (PCA) as described in our previous study⁴⁹” but I’m not sure what this means. PCA would be used to remove different ancestry, but genetic relatedness would be based on estimated IBD; is this what was used to screen for close genomic relationships? In looking at ref 49, there is a mention of using the Ge et al. (2016) method for computing heritability, but the current study states that GCTA was used, so I got a bit confused about what was actually done here. Could the authors please clarify this in the current methods section rather than referring the reader to this prior work for details? Should these h² estimates be compared to GCTA analyses of unrelated individuals? Perhaps these high estimates are to be expected with small (for whole-genome analysis) samples, esp. using GCTA (or something other than GCTA) vs. LD-score regression, and if so, it would be useful for the authors to clarify that.

The rebuttal letter explains that the authors have obtained an updating-specific Neurosynth map and

the text mentions testing whether basal ganglia connectivity predicts the updating-specific component. Thank you for adding these analyses, but I don't see the Neurosynth map presented in the relevant figure or described (generally) in the text. There is a mention of the updating-specific component in the added paragraph on p. 18, but this may seem to readers a bit out of the blue given that there is no associated updating-specific panel in Figure 5. Could the authors add it to figure 5?

p. 39: "96 trails" ◇ "96 trials"

Reviewer #3 (Remarks to the Author):

The authors have done a good job of addressing my comments and I have improved the neuroimaging aspect of this work considerably. I have signed this review for transparency. – Dustin Scheinost

REVIEWER COMMENTS

Reviewer #1 (Remarks to the Author):

Overall the authors have done a nice job of addressing the majority of my concerns.

There are a few relatively minor revisions that should be made before publication.

We appreciate this reviewer's positive evaluation of our manuscript and his/her valuable suggestions to further improve the manuscript.

1. Stats should be $p < 1 \times 10^{-3}$ rather than just 10^{-3} as in "($r = .27$, $p < 10^{-3}$, uncorrected)"

We have made the correction.

2. The information presented in lines 366-381 (associated brain regions and their MNI coordinates) would be better presented in table format if feasible.

We thank this reviewer for the helpful advice and have presented the information in table format (Table 3).

3. Please specify which GCTA tool was used to calculate the genetic correlations.

We have added this information in this revision (lines 504~507), which reads: "Second, we estimated the genetic correlations between the EF components within the same models using the Bivariate GREML functions (<https://yanglab.westlake.edu.cn/software/gcta/#BivariateGREMLanalysis>) in the GCTA toolbox (Lee, Yang, Goddard, Visscher, & Wray, 2012; Yang, Lee, Goddard, & Visscher, 2011)."

4. Related, the standard errors should be reported along with the genetic correlations and p values.

We have added this information in this revision.

5. Please specify sources for the genes related to the analyses for schizophrenia, adhd, educational attainment, etc.

We have added this information in this revision (see details in "Data availability"). Which reads:

“The sources of the GWAS summary results are as follows: schizophrenia and ADHD (<https://www.med.unc.edu/pgc/results-anddownloads/>); intelligence (<https://ctg.cncr.nl/>); educational attainment (<https://www.thessgac.org/data>); and Crohn’s disease (<https://www.ibdgenetics.org/downloads.html>). The list of genes preferentially expressed in the central nervous system was obtained from the corresponding author of the study (Lee, DeCandia, et al., 2012).”

Reviewer #2 (Remarks to the Author):

Overall, the authors have been very responsive to reviewer comments, and I believe they have addressed my comments in this revision. I have only a few minor comments/suggestions that came up in response to some of the additions:

We again thank this reviewer for acknowledging our effort and are grateful for the comments to improve the manuscript.

First, the result the SNP-h² for IQ is comparable to that for twin studies is striking, as typically the SNP-h² is ~1/2 the twin h² (doi: 10.1038/nrg.2017.104). Although the authors state that “Using the same sample, we also found higher heritability estimates of functional connectivity edges in a previous study⁴⁹ as compared to those from the Biobank data⁵⁹” (p. 22), the discrepancies there are much less pronounced, as ref 49 reports SNP-h² < 10%. (Also, I think the authors mean “the UK Biobank data”?)

Yes, we compared our results to those from the UK Biobank data. This information has been added in this revision.

In contrast, the SNP-h² for EFs and IQ are in the 50-60% range here, and I’m not sure that could be explained by sample homogeneity alone. That the SNP-h² are so high raises some concerns that something may be different in this analysis, making it less comparable to the reports in the existing literature. However, I had difficulty understanding what that might be when I re-read the methods with this in mind. The methods specify that “To make sure our participants were unrelated and had a similar genetic background (e.g., Han Chinese), we estimated the genetic relationship for each pair of participants using the principal component analysis (PCA) as described in our previous study⁴⁹” but I’m not sure what this means. PCA would be used to remove different ancestry, but genetic relatedness would be based on estimated IBD; is this what was used to screen for close genomic relationships? In looking at ref 49, there is a mention of using the Ge et al. (2016) method for computing heritability, but the current study states that GCTA was used, so I got a bit confused about what was actually done here. Could the authors please clarify this in the current methods section rather than referring the reader to this prior

work for details?

We apologize for the confusions here and have further clarified our methods in this revision (lines 1161~1171), which reads:

“To remove close relatives, we first estimated the genetic relationship for each pair of participants using GCTA functions (Yang et al., 2010; Yang et al., 2011)

(<https://yanglab.westlake.edu.cn/software/gcta/#MakingaGRMresulting>), which results in a genetic relationship matrix (GRM). Each value in the GRM reflects the average correlation of SNP values between two participants over a number of SNPs. Three pairs of participants showed estimated genetic relatedness >0.05 , so one participant from each pair was randomly selected and removed. In addition, to check the ancestry of the participants (e.g., Han Chinese), we performed the principal component analysis (PCA)(Price et al., 2006) implemented in the GCTA software (Yang et al., 2011). Nine participants were identified as non-Han Chinese and were then removed, leaving a final sample of 2110 Han Chinese participants for further analyses (Supplementary Fig. 1).”

Should these h^2 estimates be compared to GCTA analyses of unrelated individuals?

Thanks for raising this issue. We have compared our results to a previous GCTA analysis of 2875 children at age 12, which found a heritability of 0.45. We have revised this part (lines 492~502), which reads:

“We also estimated the heritability of IQ (Raven’s Progressive Matrices Test), which revealed a heritability of $h^2_{SNP} = .50$ ($SE = .38$, $p = .10$), in line with a previous GCTA analysis of 2875 children at age 12 ($h^2_{SNP} = 0.45$) (Trzaskowski, Yang, Visscher, & Plomin, 2014) and a meta-analysis based on 50 years of twin studies ($h^2 = .54$)(Polderman et al., 2015). This value is higher than that found in a prior consortium study based on unrelated individuals(Savage et al., 2018) i.e., $h^2_{SNP} = .19\sim.22$ across age groups; $h^2_{SNP} = .22$, $SE = .10$ for young adults ($n = 6033$). The higher heritability estimates may be due to the homogeneity of our sample of healthy young college students of Han ethnicity. Using the same sample, we also found higher heritability estimates of functional connectivity edges in a previous study(Feng et al., 2020) as compared to those from the UK Biobank data(Elliott et al., 2018). In addition, the limited sample size may have also led to less accurate (possibly inflated) heritability estimates.

Perhaps these high estimates are to be expected with small (for whole-genome analysis) samples, esp. using GCTA (or something other than GCTA) vs. LD-score regression, and if so, it would be useful for the authors to clarify that.

We agree with this reviewer that the higher estimates may also likely be due to the limited sample size, in addition to the homogeneity of our sample. In this revision, we have added this point (lines 501~502), which reads: “In addition, the limited sample size may have also led to less accurate (possibly inflated) heritability estimates.”

The rebuttal letter explains that the authors have obtained an updating-specific Neurosynth map and the text mentions testing whether basal ganglia connectivity predicts the updating-specific component. Thank you for adding these analyses, but I don't see the Neurosynth map presented in the relevant figure or described (generally) in the text. There is a mention of the updating-specific component in the added paragraph on p. 18, but this may seem to readers a bit out of the blue given that there is no associated updating-specific panel in Figure 5. Could the authors add it to figure 5?

We have added the Neurosynth results of updating-specific component (Figure 5E).

p. 39: “96 trails” \diamond “96 trials”

We have corrected this error.

Reviewer #3 (Remarks to the Author):

The authors have done a good job of addressing my comments and I have improved the neuroimaging aspect of this work considerably. I have signed this review for transparency. – Dustin Scheinost

We thank Prof Scheinost for acknowledging our effort and for his constructive comments to improve the manuscript.

References

- Elliott, L. T., Sharp, K., Alfaro-Almagro, F., Shi, S., Miller, K. L., Douaud, G., . . . Smith, S. M. (2018). Genome-wide association studies of brain imaging phenotypes in UK Biobank. *Nature*, *562*(7726), 210-216. doi:10.1038/s41586-018-0571-7
- Feng, J., Chen, C., Cai, Y., Ye, Z., Feng, K., Liu, J., . . . Xue, G. (2020). Partitioning heritability analyses unveil the genetic architecture of human brain multidimensional functional connectivity patterns. *Hum Brain Mapp*, *41*(12), 3305-3317. doi:10.1002/hbm.25018
- Lee, S. H., DeCandia, T. R., Ripke, S., Yang, J., Schizophrenia Psychiatric Genome-Wide Association Study, C., International Schizophrenia, C., . . . Wray, N. R. (2012). Estimating the proportion of variation in susceptibility to schizophrenia captured by common SNPs. *Nat Genet*, *44*(3), 247-250.

doi:10.1038/ng.1108

- Lee, S. H., Yang, J., Goddard, M. E., Visscher, P. M., & Wray, N. R. (2012). Estimation of pleiotropy between complex diseases using single-nucleotide polymorphism-derived genomic relationships and restricted maximum likelihood. *Bioinformatics*, *28*(19), 2540-2542. doi:10.1093/bioinformatics/bts474
- Polderman, T. J. C., Benyamin, B., de Leeuw, C. A., Sullivan, P. F., van Bochoven, A., Visscher, P. M., & Posthuma, D. (2015). Meta-analysis of the heritability of human traits based on fifty years of twin studies. *Nat Genet*, *47*(7), 702-+. doi:10.1038/ng.3285
- Price, A. L., Patterson, N. J., Plenge, R. M., Weinblatt, M. E., Shadick, N. A., & Reich, D. (2006). Principal components analysis corrects for stratification in genome-wide association studies. *Nat Genet*, *38*(8), 904-909. doi:10.1038/ng1847
- Savage, J. E., Jansen, P. R., Stringer, S., Watanabe, K., Bryois, J., de Leeuw, C. A., . . . Posthuma, D. (2018). Genome-wide association meta-analysis in 269,867 individuals identifies new genetic and functional links to intelligence. *Nat Genet*, *50*(7), 912-919. doi:10.1038/s41588-018-0152-6
- Trzaskowski, M., Yang, J., Visscher, P. M., & Plomin, R. (2014). DNA evidence for strong genetic stability and increasing heritability of intelligence from age 7 to 12. *Mol Psychiatry*, *19*(3), 380-384. doi:10.1038/mp.2012.191
- Yang, J., Benyamin, B., McEvoy, B. P., Gordon, S., Henders, A. K., Nyholt, D. R., . . . Visscher, P. M. (2010). Common SNPs explain a large proportion of the heritability for human height. *Nat Genet*, *42*(7), 565-569. doi:10.1038/ng.608
- Yang, J., Lee, S. H., Goddard, M. E., & Visscher, P. M. (2011). GCTA: a tool for genome-wide complex trait analysis. *Am J Hum Genet*, *88*(1), 76-82. doi:10.1016/j.ajhg.2010.11.011

REVIEWERS' COMMENTS

Reviewer #1 (Remarks to the Author):

I appreciate the authors' attentive and thorough response to my comments. Sincerely, Chelsie Benca-Bachman (signed for transparency)

Reviewer #2 (Remarks to the Author):

The authors have done a good job of responding to my comments and those of the other reviewers. I believe this study is an impressive piece of work and will be an important contribution to the literature.

REVIEWERS' COMMENTS

Reviewer #1 (Remarks to the Author):

I appreciate the authors' attentive and thorough response to my comments.

Sincerely, Chelsie Benca-Bachman (signed for transparency)

Thank you, Prof Benca-Bachman.

Reviewer #2 (Remarks to the Author):

The authors have done a good job of responding to my comments and those of the other reviewers. I believe this study is an impressive piece of work and will be an important contribution to the literature.

Thank you.